# Production of germ-free mosquitoes via transient colonisation allows stage-specific investigation of host–microbiota interactions

Ottavia Romoli [1✉], Johan Claes Schönbeck[1], Siegfried Hapfelmeier [2] & Mathilde Gendrin [1,3✉]

The mosquito microbiota impacts the physiology of its host and is essential for normal larval development, thereby influencing transmission of vector-borne pathogens. Germ-free mosquitoes generated with current methods show larval stunting and developmental deficits. Therefore, functional studies of the mosquito microbiota have so far mostly been limited to antibiotic treatments of emerging adults. In this study, we introduce a method to produce germ-free *Aedes aegypti* mosquitoes. It is based on reversible colonisation with bacteria genetically modified to allow complete decolonisation at any developmental stage. We show that, unlike germ-free mosquitoes previously produced using sterile diets, reversibly colonised mosquitoes show no developmental retardation and reach the same size as control adults. This allows us to uncouple the study of the microbiota in larvae and adults. In adults, we detect no impact of bacterial colonisation on mosquito fecundity or longevity. In larvae, data from our transcriptome analysis and diet supplementation experiments following decolonisation suggest that bacteria support larval development by contributing to folate biosynthesis and by enhancing energy storage. Our study establishes a tool to study the microbiota in insects and deepens our knowledge on the metabolic contribution of bacteria to mosquito development.

[1] Microbiota of Insect Vectors Group, Institut Pasteur de la Guyane, Cayenne, French Guiana, France. [2] Institute for Infectious Diseases, University of Bern, Bern, Switzerland. [3] Parasites and Insect Vectors Department, Institut Pasteur, Paris, France. ✉email: oromoli@pasteur-cayenne.fr; mathilde.gendrin@pasteur.fr

Vector-borne diseases are estimated to account for 17% of all cases of infectious diseases worldwide and cause over 700,000 deaths per year[1]. The mosquito *Aedes aegypti* is the main vector for several arboviruses, and it is widely distributed across tropical and neotropical regions. Due to the spread of insecticide resistance, the factors shaping mosquito fitness and vector competence (i.e. its permissiveness to virus development) are under study in order to identify alternative strategies for transmission blockade. Among these, the microbial community colonising the mosquito is known to directly impact vector competence and other life-history traits affecting transmission success, including mosquito development, lifespan, mating choice, fecundity and fertility[2–7]. In particular, live bacteria are essential for normal larval development, but the exact contribution of the microbiota to larval development is still a conundrum. It has been proposed that the bacteria growing in the mosquito gut consume oxygen and that the resulting hypoxia is a developmental signal for moulting from one stage to the next[8]. However, successful larval development in fully germ-free conditions challenged this model and metabolic contribution of bacteria to larval development was alternatively hypothesised[9].

Many aspects of adult mosquito physiology are determined by larval development. Different parameters such as larval density, temperature and nutrition directly impact larval development and indirectly control different adult traits such as size, fecundity and lifespan[10–13]. Moreover, the presence of a larval microbiota and its composition have been shown to affect the adult size and physiology. More specifically, mosquitoes produced in the absence of a microbiota are smaller than colonised mosquitoes[9], and mosquitoes carrying different strains of bacteria during development show differences in adult size and vector competence even if they carry similar microbiota during adulthood[10].

Considering such larvae-to-adult carry-over effects, we felt that tools were missing to specifically assess the role of the microbiota in larvae and in adults. The current gold standard to avoid any carry-over effect is the use of antibiotics at adult emergence after conventional larval rearing. However, antibiotics may have off-target effects (notably on mitochondria) and do not fully sterilise mosquitoes[14]. In the past, several studies proposed rearing protocols of germ-free larvae[15,16], which could not be reproduced in recent experiments[9]. Lately, mosquito larvae have been shown to develop in the absence of bacteria when provided an enriched diet embedded in agar plugs rather than food in suspension, but development time is tripled and resultant germ-free mosquitoes are smaller than their conventional controls[9].

We developed a transient bacterial colonisation method to support larval development while producing fully developed germ-free adults. Using this approach, we were able to support development within a standard timeline and without affecting adult size, showing that these germ-free individuals represent a valid tool for mosquito microbiota studies. We used this approach to assess the role of the microbiota in larvae and adults. In adults, we did not detect any impact of bacterial colonisation on the proportion of mosquitoes laying eggs, the egg clutch size or the mosquito lifespan. In larvae, we used decolonisation in the middle of larval development to characterise the role of bacteria using a transcriptomic approach and diet complementation. Our data suggest that bacteria contribute to folate biosynthesis and energy storage and that folic acid supports larval development in the absence of a microbiota.

## Results

### Production of germ-free adult mosquitoes via transient bacterial colonisation.
After microbiological sterilisation of mosquito eggs, hatchlings provided with autoclaved standard rearing diet are able to survive for weeks but remain first-instar larvae. Their development can be rescued by bacterial colonisation, for instance with *Escherichia coli*[4,9]. Therefore, we hypothesised that transient bacterial colonisation would be an efficient way to support larval development and produce germ-free adults. To test this, we used HA416, an *E. coli* HS-derived strain that is auxotrophic for two non-standard amino acids required for peptidoglycan synthesis (*meso*-diaminopimelic acid, *m*-DAP, and D-alanine, D-Ala) and that was previously used to transiently colonise the gut of germ-free mice[17]. We supplemented the food of gnotobiotic larvae with *m*-DAP and D-Ala to support the proliferation of auxotrophic *E. coli* during larval development only (Fig. 1a). These supplements were toxic to first-instar larvae when provided at the concentrations previously used in vitro, but 4-fold lower concentrations supported growth of auxotrophic *E. coli* (Fig. S1) without causing toxicity in larvae. The auxotrophic *E. coli* strain (hereafter referred to as AUX) rescued the development of axenic larvae with the same efficiency as control wild-type *E. coli* (hereafter referred to as WT, $p = 0.87$, Fig. 1b). The proportion of larvae reaching adulthood was even slightly higher among gnotobiotic larvae than among untreated larvae (AUX: + 16%, WT: + 12%; $p < 0.001$; Fig. 1b and Table S1 for statistical information).

To monitor colonisation efficiency, we quantified bacterial loads at different time-points of larval and pupal development in AUX- and WT-carrying larvae (Fig. 1c). After surface sterilisation, larval and pupal homogenates were plated on lysogeny broth (LB) (supplemented with *m*-DAP and D-Ala for AUX). Comparable bacterial amounts were found in third-instar larvae harbouring each *E. coli* strain ($p = 0.71$). Bacterial counts then gradually decreased in AUX hosts and were 1.3- and 8-fold lower than in WT controls on the first and second day of the fourth instar, respectively (day 4: $p = 0.033$; day 5: $p < 0.001$). During the pupal stage, bacteria were undetected in most AUX-carrying individuals ($92.1 \pm 7.7\%$) while bacterial load also decreased over 100-fold in WT controls, as previously documented[18] ($p < 0.001$).

We monitored *E. coli* survival and/or growth in larval water. In the absence of larvae, colony forming units (CFU) count significantly decreased for both bacteria (Fig. S2a: WT: 60–73% decrease, $p = 0.0022$; AUX: 72–78% decrease, $p = 0.0064$). This is not surprising as the initial bacterial suspension is almost as concentrated as an overnight culture, and bacteria limit their proliferation in response to high cell densities via quorum sensing. AUX counts were similar to WT counts at each day except day 5 (day 5: 34% reduction, $p = 0.017$; other days: $p > 0.18$). In the presence of larvae, bacterial loads in the suspension for both strains decreased by 10-folds at day 4, when larvae had reached the fourth instar, i.e. when body size increases most significantly[19] (Fig. S2b: WT: $p < 0.001$, least square means with Bonferroni correction (lsmeans) day 3 vs day 4: $p = 0.015$; AUX: $p < 0.001$, lsmeans day 3 vs day 4: $p = 0.0074$). AUX loads in the suspension were 3-fold and $10^6$-fold lower than WT counts at days 5 and 6, respectively, in parallel with the end of the fourth instar and the beginning of the metamorphosis (day 5: $p < 0.001$; day 6: $p < 0.001$). Hence, AUX concentrations in the water and in larvae point to similar dynamics.

No bacteria could be cultured from adults reared on AUX *E. coli* during the first 3 days following emergence, while bacteria were culturable from 50% of the WT *E. coli* gnotobiotic mosquitoes at the same time-points (Fig. 1d). The absence of culturable bacteria in some conventionally reared mosquitoes immediately after adult emergence has already been reported in different mosquito species, including *Ae. aegypti*[18]. To further quantify the loss of bacteria, we measured the bacterial loads in adult mosquitoes originating from auxotroph-carrying gnotobiotic larvae on a larger sample size. No culturable bacteria was

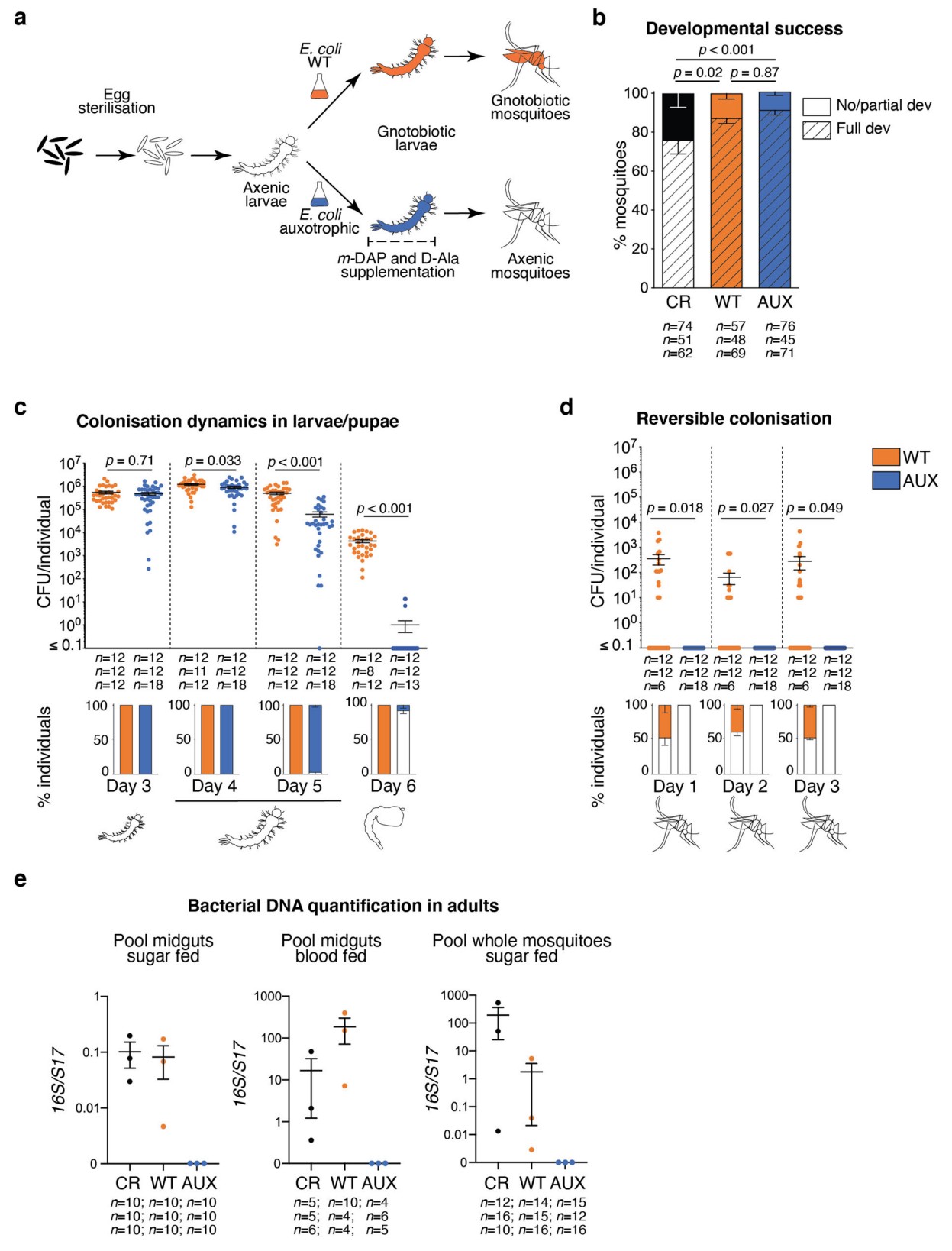

isolated from 97.6 ± 2.1% (34/34; 118/123; 311/321) of 0–2 day old adults (Fig. S3). Bacterial DNA was not detectable via qPCR in 7–9 day old mosquitoes obtained from AUX larvae, whether DNA was extracted from pools of sugar-fed or blood-fed midguts or from whole mosquitoes (Fig. 1e).

**Development dynamics on auxotrophic and wild-type bacteria.** We investigated whether larvae harbouring AUX *E. coli* presented any developmental deficits. Firstly, we quantified the duration of each developmental stage to assess if the colonising bacterial strains had any impact on development kinetics. The gnotobiotic

**Fig. 1 Reversible colonisation allows the production of germ-free mosquitoes. a** Schematic representation of the experimental setup. Eggs are surface sterilised to give rise to axenic larvae. A bacterial culture of the *E. coli* wild-type (gnotobiotic control) or auxotrophic strain is added to axenic larvae to obtain gnotobiotic larvae. To support the growth of auxotrophic bacteria, *meso*-diaminopimelic acid (*m*-DAP) and D-alanine (D-Ala) are provided throughout the whole larval developmental stage. After transferring pupae carrying auxotrophic *E. coli* to a new medium lacking *m*-DAP and D-Ala, germ-free mosquitoes are obtained. **b** Proportion of larvae developing to adulthood (striped bars) and of larvae dead or stalled in the development (full bars) after conventional rearing (CR, black), rearing with wild-type *E. coli* (WT, orange) and auxotrophic *E. coli* (AUX, blue). CR larvae developed from non-sterilised eggs in tap water on autoclaved fish food. **c** Colony forming units (CFU) quantification of bacterial loads (upper chart) and prevalence (lower chart) in larvae at third instar (Day 3), early fourth instar (Day 4), late fourth instar (Day 5) and in pupae (Day 6). **d** CFU quantification of bacterial loads (upper chart) and prevalence (lower chart) in 0–3 day old mosquitoes. **e** Detection of bacterial DNA via qPCR in midguts or whole body of 7–9 day old adult mosquitoes. 0 reflects non-detection or detection below the negative control. In (**b**–**d**), data show mean ± SEM of three independent replicates. The exact number of individuals analysed per condition and replicate is indicated in each panel. In (**e**) data show mean ± SEM of three independent replicates. The number of individuals pooled in each replicate is indicated in the figure. Statistical significance was determined with generalised linear mixed models and least square means with Bonferroni correction. Exact *p* values are indicated in the figure. See Table S1 for detailed statistical information. Source Data are provided as a Source Data file.

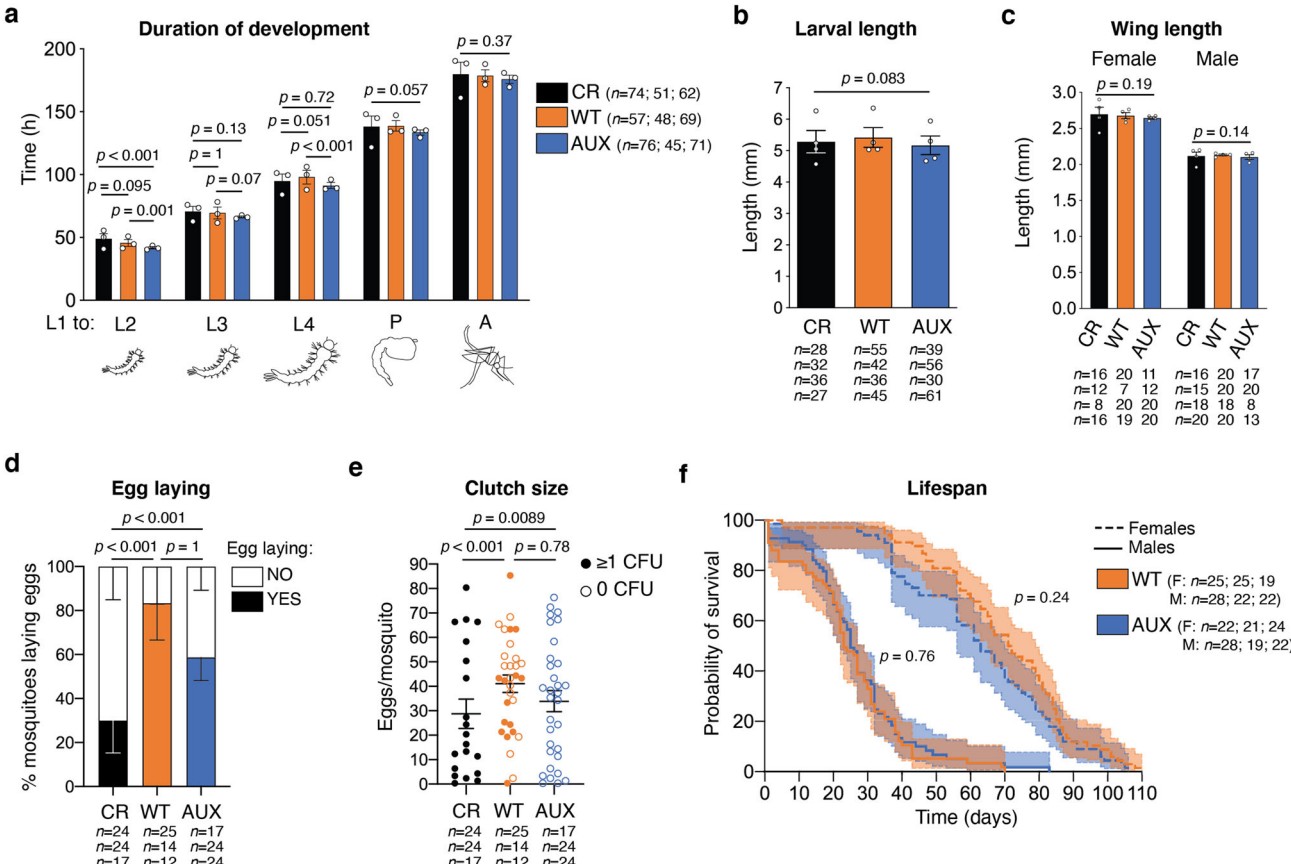

**Fig. 2 Auxotrophic bacteria support mosquito growth without any detected deficit in development, fecundity or lifespan. a** Duration of the larval instars and of the pupal stage after conventional rearing (CR, black), rearing with wild-type *E. coli* (WT, orange) and auxotrophic *E. coli* (AUX, blue). The duration of the different stages is quantified from the addition of bacteria to first-instar larvae (L1) to the beginning of each stage. **b** Larval length measured on the fifth day after hatching. **c** Wing length of 2 day old adult mosquitoes. **d** Proportion of mosquitoes laying eggs (full bars) after the first blood meal. **e** Number of eggs laid per mosquito after the first blood meal. Full circles represent mosquito contaminated by bacteria (colony forming units, CFU ≥ 1) while empty circles represent germ-free mosquitoes (0 CFU). **f** Kaplan-Meier survival curves of male (solid line) and female (dotted line) sugar-fed mosquitoes. In (**a**–**e**), data show mean ± SEM of three (**a**, **d**, **e**) or four (**b**, **c**) independent replicates. The exact number of individuals analysed per condition and replicate is indicated in each panel. Statistical significance was determined with generalised linear mixed models and least square means with Bonferroni correction. In (**f**), data show survival proportions ± 95% confidence interval of three replicates. The exact number of individuals analysed per condition and replicate is indicated in the figure. Statistical significance was determined with a Cox proportional hazards model. Exact *p* values are indicated in the figure. See Table S1 for detailed statistical information. Source Data are provided as a Source Data file.

treatments slightly shortened the beginning of larval development, while total development time was unaffected (Fig. 2a, L1 to L2: $p < 0.001$; to L3: $p = 0.044$; to L4: $p = 0.001$; to pupa: $p = 0.057$; to adult: $p = 0.37$). Secondly, we evaluated whether AUX *E. coli* supported the development of both sexes. A 16% increase

in the proportion of females was observed in WT *E. coli*-carrying mosquitoes compared to conventionally reared individuals, but differences in sex ratio between all three conditions were not statistically significant (Fig. S4, $p = 0.09$). Thirdly, we compared the larval body length and the wing length of conventionally

reared and gnotobiotic individuals to assess whether the different microbiota compositions impacted the mosquito size, which is a readout of the insect nutritional and metabolic status. At the fifth day after egg hatching, gnotobiotic fourth-instar larvae for both *E. coli* strains reached the same length as conventionally reared larvae (Fig. 2b, $p = 0.083$) and wing size was comparable among the three rearing conditions in both males and females (Fig. 2c, females: $p = 0.19$; males: $p = 0.14$).

**Reproduction success and lifespan of E. coli-colonised and germ-free adults**. We took advantage of our experimental setup to investigate the importance of bacterial colonisation during adulthood on fertility and lifespan, independent of the effects of the larval microbiota. Results showed that a higher proportion of gnotobiotic individuals laid eggs compared to conventionally reared mosquitoes, and that the size of their egg clutches was larger (Fig. 2d: $p < 0.001$; Fig. 2e: $p < 0.001$). WT and germ-free mosquitoes had similar fecundity with a similar proportion of egg-laying females ($p = 1.0$) and similar clutch sizes ($p = 0.78$). This was further confirmed when analysing egg production in WT females only, which was similar whether or not they carried a detectable amount of bacteria (egg clutch size: $p = 0.22$; proportion of mosquitoes laying eggs: $p = 0.74$). No difference in the proportion of egg clutches producing viable larvae was detected among all conditions (Fig. S5: $p = 1.0$).

Finally, we investigated the effect of *E. coli* colonisation on adult lifespan. Pupae originating from AUX and WT *E. coli* larvae were rinsed in sterile water, sorted by sex, and placed in sterile boxes for adult rearing along with a sterile sucrose solution. Both female and male lifespan was not significantly influenced by *E. coli* colonisation (Fig. 2f, Cox proportional hazards model females: $p = 0.24$; males $p = 0.76$).

**Transient colonisation allows the production of germ-free larvae**. We tested whether reversible colonisation could also be arrested during larval development by placing AUX-harbouring larvae in a rearing medium lacking *m*-DAP and D-Ala (Fig. 3a). For that purpose, we rinsed third-instar larvae and placed them in sterile water containing non-supplemented sterile fish food and monitored the bacterial loads in single larvae. As soon as 2 h after *m*-DAP and D-Ala depletion, we observed a 500-fold reduction in bacterial load with respect to non-transferred larvae, which became a $10^4$-fold reduction after 5 h; 95% of larvae were germ-free 12 h after amino-acid depletion (Fig. 3b, lsmeans: Not transferred vs transferred AUX at each time-point: $p < 0.001$). We processed WT *E. coli*-carrying larvae in the same way to test whether this reduction in the microbiota load was specific to the bacterial auxotrophy or if it was only due to the transfer of larvae to a new sterile rearing environment and the active expulsion of bacteria from the larval gut. Two hours after the transfer, we observed a 300-fold reduced bacterial load, suggesting that the change of medium has a substantial effect on the microbiota load. However, the bacterial load of transferred larvae started to increase between 2 and 5 h after the transfer and after 20 h it was similar to non-transferred controls (Fig. 3b, WT: $p = 0.12$). We tested whether, as already known for first-instar larvae, late larval development was affected by the loss of bacteria. When early third-instar larvae were set to turn germ-free, they all moulted to the fourth instar ($99 \pm 0.49\%$, mean $\pm$ SEM), but only $30 \pm 6.2\%$ moulted to pupa and only $23 \pm 4.6\%$ fully developed into adults (Fig. 3c). A significant reduction in the developmental success after transfer was also observed in WT-carrying larvae, with only $43 \pm 5.4\%$ of individuals reaching adulthood (Fig. 3c, L4: $p = 1.0$; pupae: $p < 0.001$; adults: $p < 0.001$, lsmeans on adults: all

comparisons: $p < 0.001$). Larval development was significantly longer in both germ-free and WT transferred larvae (Fig. S6, $p < 0.001$), with the fourth instar duration being the most affected in germ-free larvae (L3: $p < 0.001$; L4: $p < 0.001$; metamorphosis: $p = 0.04$).

We verified if the colonisation could be interrupted at other larval stages, by transferring AUX-carrying larvae into sterile water lacking *m*-DAP and D-Ala every 24 h for 96 h. On average, $89 \pm 11\%$ of larvae completely lost their microbiota after 16 h from transfer, revealing that the colonisation with AUX bacteria can efficiently be switched-off at any stage of larval development (Fig. 3d). When we investigated the development success of larvae decolonised at different stages, we observed that the percentage of individuals completing their development positively correlates with the duration of colonisation, varying from $2.5 \pm 1.5\%$ for larvae transferred after 24 h to $92 \pm 3.2\%$ for larvae transferred after 96 h, i.e. during their fourth instar (Fig. 3e). Larval development was 50% longer in individuals decolonised during early time-points (24 h: $9.2 \pm 0.2$ days; 48 h: $9.3 \pm 0.7$ days, Fig. S7) compared to mosquitoes decolonised at later instars (72 h: $6.2 \pm 0.2$ days; 96 h: $5.9 \pm 0.1$ days, Fig. S7). Considering WT-carrying larvae, it appeared that the transfer into a sterile environment had a more significant impact on second/third-instar larvae (48 h), as those individuals showed the lowest developmental success ($56 \pm 12\%$, lsmeans 48 h vs all time-points: $p < 0.001$, Fig. 3e) and a longer development time ($7.0 \pm 0.5$ days, Fig. S7). Larval development was also longer in larvae transferred after 24 h ($7.0 \pm 0.5$ days, Fig. S7), but those individuals showed developmental levels comparable to larvae transferred at later time-points ($83 \pm 7.5\%$, lsmeans 24 h vs 72 h: $p = 1$; 24 h vs 96 h: $p = 0.95$, Fig. 3e). Taken together, these observations confirm that bacterial contribution is essential for development, especially to the first half of larval development.

**Impact of the transfer of larvae carrying a native microbiota**. Since the *E. coli* HS strain used in these experiments is a human commensal, we asked whether the developmental deficit observed after transferring the larvae into sterile water was due to a mis-adaptation of this *E. coli* strain to the mosquito gut. After egg sterilisation, germ-free larvae were reared in water collected from *Ae. aegypti* breeding sites in Cayenne (French Guiana). Since developmental synchronisation was more variable among these larvae, we investigated the effect of transferring larvae into a sterile environment at two different time-points, i.e. 48 and 72 h after colonisation, corresponding to larval populations enriched in second and third instars or third and fourth instars, respectively. The earlier transfer at 48 h induced a 130-fold decrease in larval CFU counts 2 h after transfer, and a further 6-fold decrease after 3 additional hours (Fig. 4a, $p < 0.001$). When transferred 72 h after colonisation, larvae showed a 10-fold reduction in bacterial counts 2 h after transfer, and a further 8-fold decrease after 3 h (Fig. 4b, $p < 0.001$). Similar to results obtained with WT *E. coli*, bacteria grew again at later time-points and no significant difference was measured in CFU counts 24 h after transfer for both time-points (48 h: $p = 0.36$; 72 h: $p = 0.18$). Again, we observed that development success decreased upon larval transfer (Fig. 4c, 48 h: non-transferred $84 \pm 15\%$, transferred $63 \pm 13\%$, $p < 0.001$; Fig. 4d, 72 h: non-transferred $88 \pm 6\%$, transferred $77 \pm 7\%$, $p = 0.002$). Surprisingly, we also observed a higher mortality shortly after larval transfer instead of the development blockade observed among gnotobiotic larvae (Fig. 4c, 48 h: non-transferred $5 \pm 5\%$; transferred $34 \pm 13\%$, $p = 0.0081$; Fig. 4d, 72 h: non-transferred $4 \pm 2\%$; transferred $20 \pm 7\%$, $p < 0.001$). Together, these data suggest that the transfer into a sterile environment also impacts the gut colonisation of a more conventional microbiota

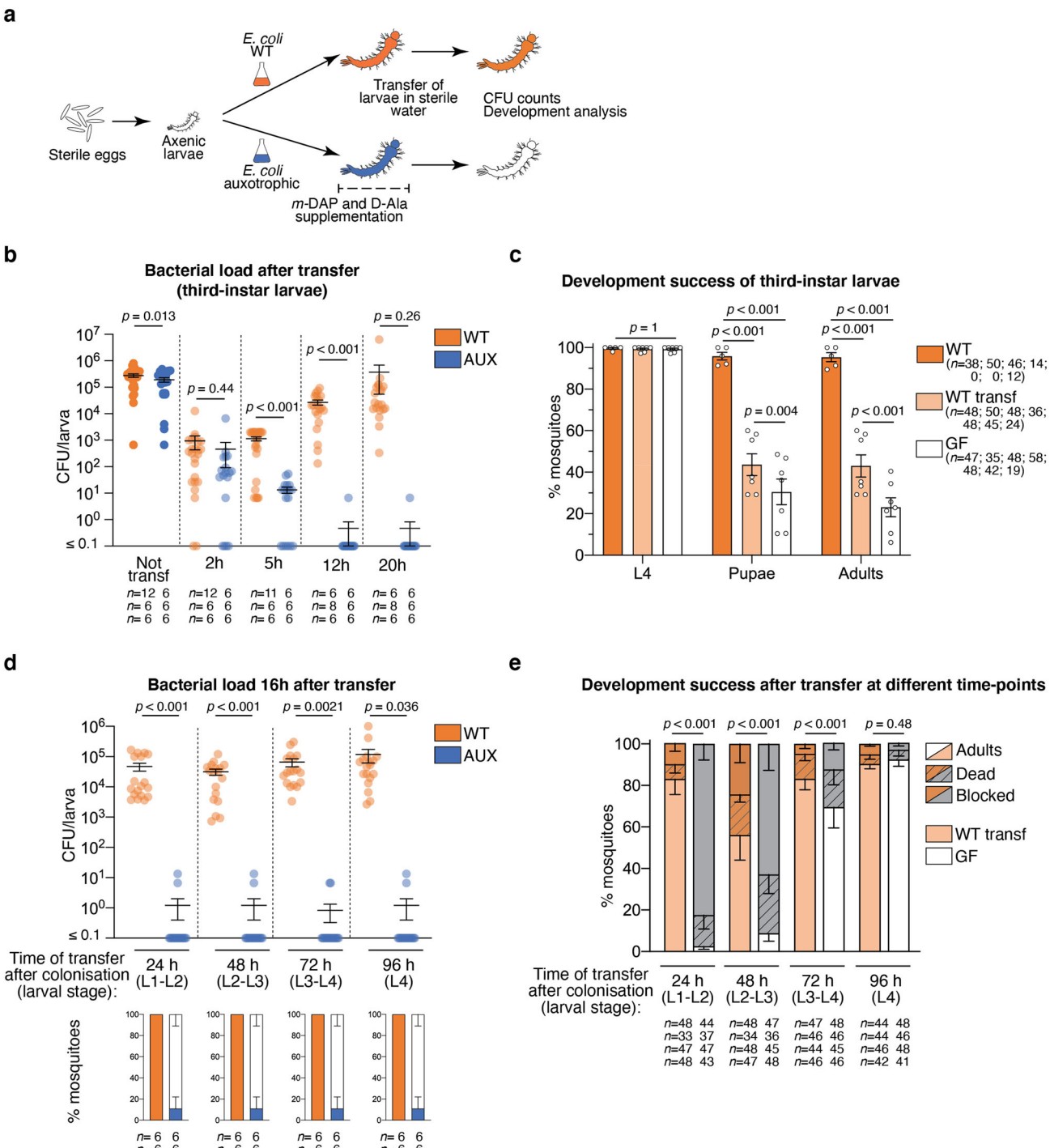

**Fig. 3 Decolonisation dynamics and developmental analysis of germ-free larvae obtained via reversible colonisation. a** Schematic representation of the experimental setup. To stop bacterial colonisation, larvae are transferred to a sterile rearing water lacking *meso*-diaminopimelic acid (*m*-DAP) and D-alanine (D-Ala). Bacterial loads are measured via colony forming units (CFU) count at different time-points and the development success of transferred larvae is assessed. **b** Bacterial loads of third-instar larvae gnotobiotic for wild-type (WT, orange) or auxotrophic (AUX, blue) *E. coli* after 2, 5, 12 and 20 h from transferring into sterile rearing water. **c** Development success of larvae continuously colonised (WT, orange), reared on wild-type (WT transf, light orange) or auxotrophic (GF, white) *E. coli* until the third instar and transferred into sterile rearing water. The proportion of individuals reaching the fourth instar (L4), the pupal stage and adulthood is shown. **d** Bacterial loads of larvae gnotobiotic for wild-type (WT, orange) or auxotrophic (AUX, blue) *E. coli* transferred into sterile rearing water at different developmental stages (L1: first instar, L2: second instar, L3: third instar, L4: fourth instar). CFU counts were measured 16 h after transfer. **e** Proportion of individuals reaching adulthood (full bar), dying (striped dark bars) and with stalled development (dark bars) after being reared with wild-type (WT transf, light orange) or auxotrophic (GF, white) *E. coli* and being transferred into sterile rearing water after 24, 48, 72 or 96 h since colonisation. In (**b–e**), data show mean ± SEM of three (**b, d**), five (**c**, WT), seven (**c**, WT transferred and GF) or four (**e**) independent replicates. The exact number of individuals analysed per condition and replicate is indicated in each panel. Statistical significance was determined with generalised linear mixed models and least square means with Bonferroni correction. Exact *p* values are indicated in the figure. See Table S1 for detailed statistical information. Source Data are provided as a Source Data file.

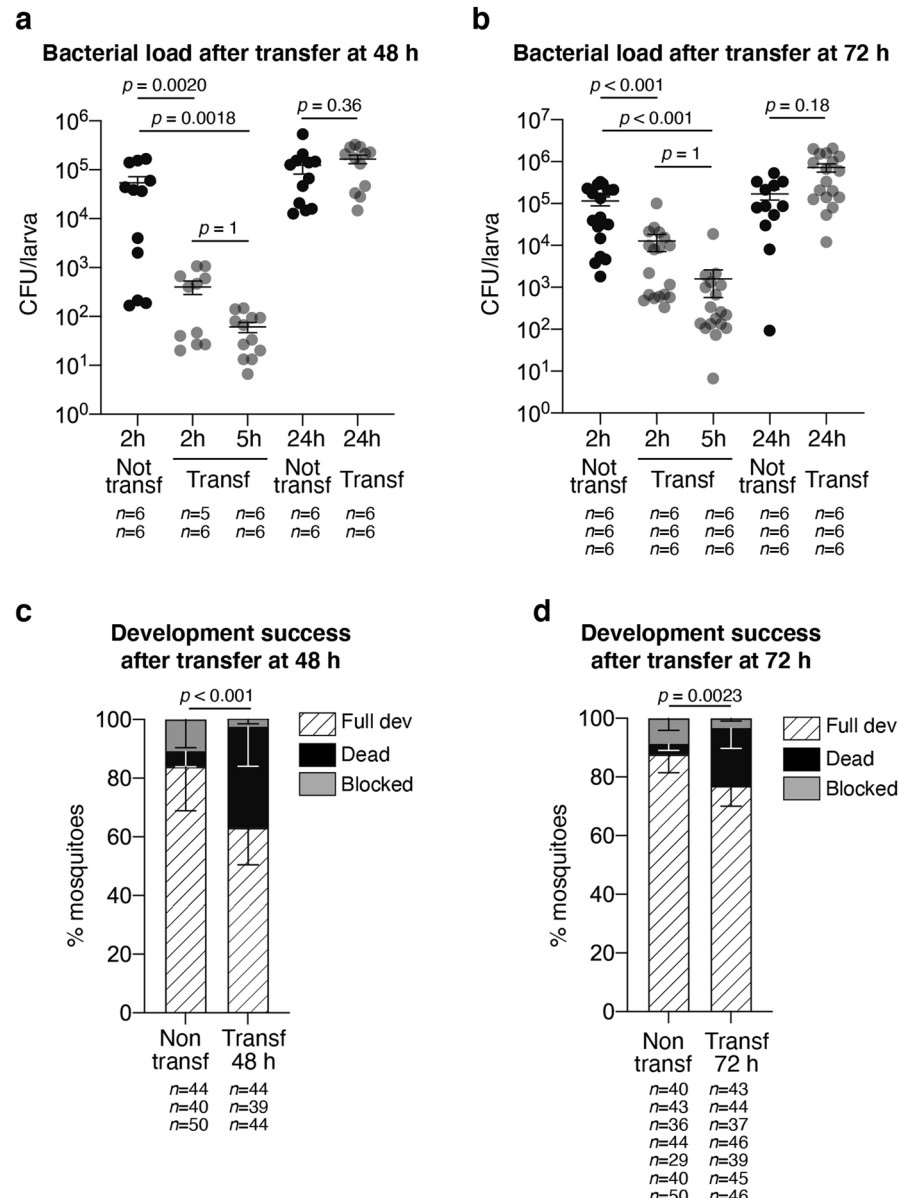

**Fig. 4 Bacterial dynamics and developmental analysis of larvae reared in breeding site water and transferred to sterile rearing medium. a, b** Colony forming units (CFU) count in larvae reared in water collected from *Ae. aegypti* breeding sites for 48 h (**a**) or 72 h (**b**) after 2, 5 and 24 h from transferring into sterile rearing water. Bacterial loads of non-transferred larvae are shown in black, while those of transferred larvae are shown in grey. **c, d** Development success of larvae continuously reared in breeding site water or transferred into sterile rearing water after 48 h (**c**) or 72 h (**d**) expressed as percentage of individuals reaching adulthood (striped bars), dying (black bars) and with stalled development (grey bars). In (**a–d**), data show mean ± SEM of two (**a**), three (**b**, **c**) or seven (**d**) independent replicates. The exact number of individuals analysed per condition and replicate is indicated in each panel. Statistical significance was determined with generalised linear mixed models and least square means with Bonferroni correction. Exact *p* values are indicated in the figure. See Table S1 for detailed statistical information. Source Data are provided as a Source Data file.

and that such perturbation has a detrimental effect to larval development, whether this microbiota is composed of a single bacterial type or of a community of bacteria.

**Microbiota-dependent gene regulation during third-instar larval development.** Since AUX *E. coli* allows an efficient decolonisation of mosquito larvae, we used it to investigate the contribution of the microbiota to larval development by transcriptomic analysis. We focused on third-instar larvae because their size allows precise dissections and because this developmental stage is intermediate between two larval stages, and

therefore more representative of larval development than fourth-instar larvae which moult into pupae.

We compared the transcriptome between third-instar larvae following clearance of AUX *E. coli* and individuals carrying WT *E. coli* (referred to as germ-free and colonised, respectively). We synchronised larvae by selecting individuals starting their third instar within a 5 h timeframe, prior to transferring AUX-carrying larvae for *m*-DAP and D-Ala depletion (Fig. 5a). We sampled guts and whole larvae from germ-free and colonised conditions 12 and 20 h later, in order to detect early changes caused by the absence of bacteria prior to potential developmental delay as well as later changes caused by bacterial loss

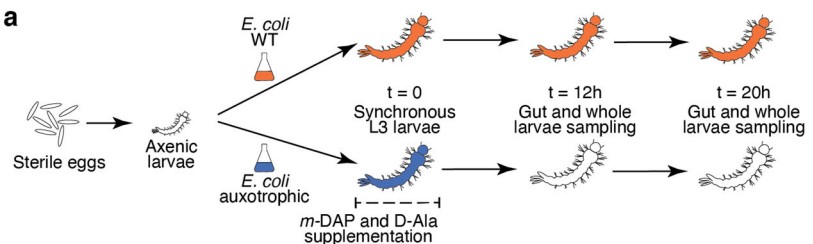

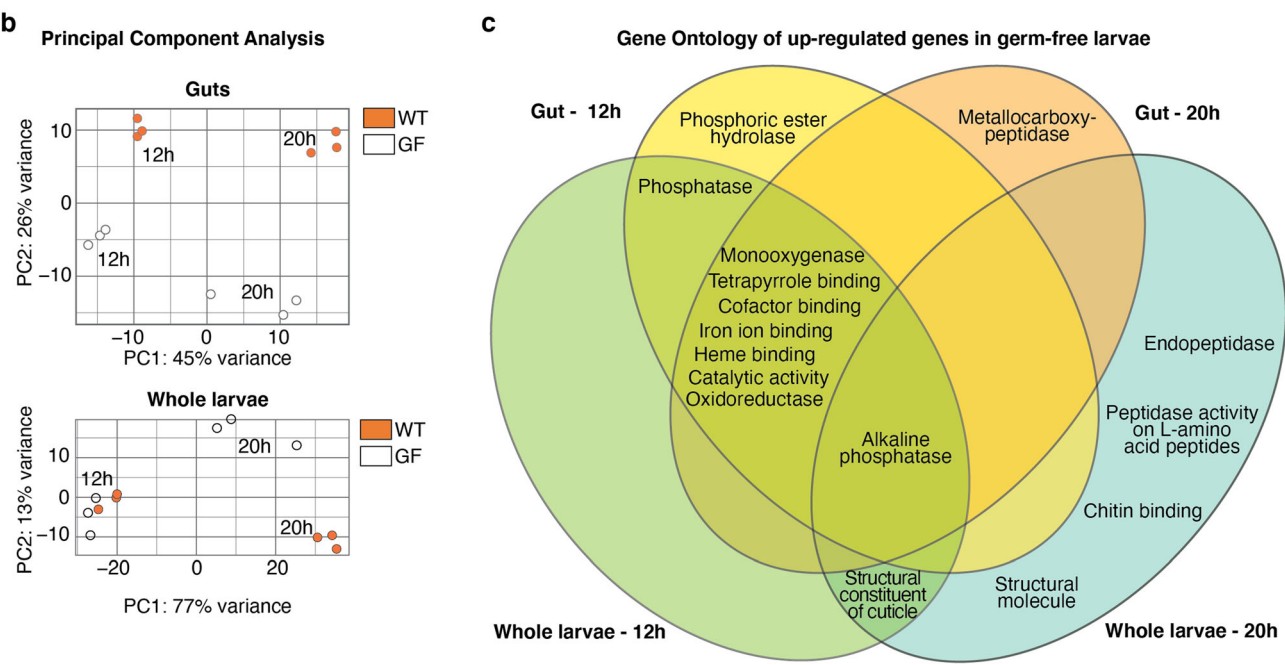

| AAEL | Description | | Function | Gut 12h | Gut 20h | Wl 12h | Wl 20h |
|---|---|---|---|---|---|---|---|
| 000931 | Alkaline phosphatase | | Phosphatase activity | 9.2 | 7.2 | 5.5 | 5.6 |
| 003317 | Alkaline phosphatase | | Phosphatase activity | 7.1 | 6.1 | 7.6 | 6.0 |
| 003297 | Alkaline phosphatase | | Phosphatase activity | 5.8 | 3.2 | 5.4 | 3.2 |
| 002555 | Sodium/solute symporter | | Transmembrane transport | 5.7 | 4.5 | 3.6 | 3.1 |
| 002138 | Triacylglycerol lipase | | Hydrolytic reaction; lipid metabolism | 4.4 | 3.9 | 4.0 | 4.0 |
| 003289 | Alkaline phosphatase | | Phosphatase activity | 5.2 | 3.0 | 4.7 | 3.1 |
| 000323 | Cysteine-rich venom protein | | Trypsine inhibitor activity | 4.0 | 3.9 | 3.7 | 3.6 |
| 029062 | Unknown (Chitin binding) | | Chitin binding | 5.0 | 3.0 | 4.5 | 2.7 |
| 024870 | Nose resistant to fluoxetine protein 6-like | | Transferase activity | 3.3 | 3.1 | 3.8 | 3.1 |
| 003905 | Alkaline phosphatase | | Phosphatase activity | 4.5 | 2.5 | 3.1 | 3.1 |

| AAEL | Description | | Function | Gut 12h | Gut 20h | Wl 12h | Wl 20h |
|---|---|---|---|---|---|---|---|
| 008045 | Hexamerin 2 beta | | Amino acid storage | - | -6.7 | -5.3 | -7.0 |
| 011169 | Hexamerin 2 beta | | Amino acid storage | -3.6 | -4.2 | -2.5 | -4.2 |
| 011520 | Sucrose transport protein | | Sucrose transport | -4.5 | -3.8 | -3.1 | -2.9 |
| 008817 | Hexamerin 2 beta | | Amino acid storage | - | -4.0 | -3.1 | -6.0 |
| 018240 | Unknown (Fuseless region) | | Pre-synaptic calcium channel regulation | -3.3 | -3.4 | -3.3 | -3.0 |
| 008313 | Unknown | | | -2.9 | -3.1 | -4.5 | -4.5 |
| 013757 | Hexamerin 2 beta | | Amino acid storage | - | -5.4 | -2.2 | -4.5 |
| 011661 | Unknown (Cilia structure/activity) | | Cilia structure/activity | -3.9 | -4.1 | -1.9 | -3.2 |
| 023132 | Unknown (Lipase activity) | | Hydrolytic reaction | -2.7 | -4.0 | -2.5 | -4.5 |
| 008598 | Unknown (Vitellogenin domain) | | Lipid metabolism | - | -2.7 | -2.0 | -4.7 |

throughout the third instar. After extraction of total RNA, we confirmed the loss of bacteria in reversibly colonised mosquitoes by qPCR (Fig. S8, gut 12 h: $p = 0.011$; gut 20 h: $p < 0.001$; whole larvae 12 h: $p < 0.001$, whole larvae 20 h: $p < 0.001$). cDNA libraries were produced and sequenced on an Illumina NextSeq

500. A total of 2735 transcripts were significantly differentially regulated (adjusted $p < 0.001$) in at least one sample type at one time-point. The sample distance heatmap showed that for each tissue type and time-point, the three experimental replicates clustered together (Fig. S9). The principal component analysis

**Fig. 5 Transcriptomic analysis of germ-free and colonised larvae. a** Schematic representation of the experimental setup. To produce third-instar germ-free larvae, larvae gnotobiotic for auxotrophic *E. coli* were transferred to a rearing medium lacking *meso*-diaminopimelic acid (*m*-DAP) and D-alanine (D-Ala). Midguts (*n* = 60/replicate) and whole larvae (*n* = 60/replicate) were sampled for RNA extraction 12 and 20 h after transfer, using wild-type *E. coli* gnotobiotic larvae as a control. Three independent replicates were performed. **b** Principal component analysis (PCA) of guts and whole larvae transcriptomic data of wild-type *E. coli*-carrying (WT, orange) and germ-free larvae (GF, white). **c** Venn diagram of Gene Ontology terms enriched for genes up-regulated in germ-free conditions. **d** $\log_2$-fold values and function of the ten highest up-regulated (red) and down-regulated (blue) genes in germ-free larvae, ordered by decreasing (up-regulated) or increasing (down-regulated) mean value in gut and whole larvae (Wl) sample (taking into account values with a *p*adj < 0.001). Significant expression values (*p*adj < 0.001) are coloured with a blue/red colour code corresponding to down/up-regulation. Hyphens mean absence from the DESeq2 output of genes with *p*adj < 0.1, meaning that transcripts are either not detected or not differentially regulated at this 0.1 threshold. To improve visibility, different colour codes indicate genes with a shared function (orange: phosphatase activity; green: hydrolytic reaction; yellow: chitin binding; purple: transferase activity; light blue: amino-acid storage; dark blue: lipid metabolism). Source Data are provided as a Source Data file.

(PCA) on gut samples identified a first component explaining 45% of the variance and discriminating time-points and a second component representing 26% of the variance and separating samples by the presence/absence of bacteria (Fig. 5b). PCA on whole larvae data revealed a more significant separation of samples by time-point (77% of the variance), with only 13% of the variance explained by the microbiota composition. The 12 h time-point was early enough to detect early transcriptional modifications in the gut before any body-scale transcriptional change, which was detectable 20 h post amino-acid depletion (Fig. 5b).

Among the differentially regulated genes, we selected the transcripts with a $\log_2$-fold value < −1.5 or > 1.5 (i.e. >4.5-fold down/up-regulation), obtaining a total of 573 up-regulated and 293 down-regulated genes in germ-free larvae compared to colonised larvae (Table S2). The Gene Ontology enrichment analysis of transcripts up-regulated in germ-free larvae included phosphatase/alkaline phosphatase/phosphoric ester hydrolase activity, oxidoreductase activity, haem binding, iron ion binding, monooxygenase activity, cofactor binding and catalytic activity (Fig. 5c and Table S3). Five alkaline phosphatases were found amongst the most highly up-regulated genes (Fig. 5d and Table S4). The Kyoto Encyclopaedia of Gene and Genomes (KEGG) enrichment analysis revealed that folate (vitamin B9) biosynthesis and thiamine (vitamin B1) metabolism were up-regulated in germ-free larvae in both gut and whole larva samples (Table S3). Focusing on the genes down-regulated in germ-free larvae, we detected 6 *hexamerin 2 beta* genes as significantly down-regulated, comprising 4 of the 10 most down-regulated genes (Fig. 5d and Table S5). Hexamerins are proteins reported to participate in amino-acid storage during the end of larval development and metamorphosis[20]. Interestingly, we also identified several down-regulated genes encoding proteins involved in lipid storage and transport, such as *vitellogenin*, a vitellogenin-related gene (*AAEL008598*, Fig. 5d and Table S5) and two *lipophorin* genes (*AAEL009955*, $\log_2$ fold −1.01 in larvae 20 h and *AAEL018219*, −0.8 in guts 20 h).

As hypoxia-induced factors (HIF) proteins are induced in the presence of a microbiota in first-instar larvae[21] and are targets of prolyl hydroxylases (PHDs), we specifically checked the expression level of *HIF-alpha* and *-beta* (*AAEL019499* and *AAEL010343*) and of *prolyl hydroxylase (PHD)-1* and *-2* genes (*AAEL024908* and *AAEL028025*). Among these genes, only *phd-2* was slightly down-regulated at 20 h in germ-free conditions (gut: $\log_2$ fold = −1.4, *p*adj = 0.0015; whole larvae: $\log_2$ fold = −0.6, *p*adj = 0.015). We also measured the oxygen levels in larval guts 12 h post transfer, which should correspond to the lowest level of oxygen concentration in the presence of bacteria[8]. No difference in oxygen levels was detected between colonised larvae (left in their medium or transferred in new sterile medium) and germ-free larvae (Fig. S10, *p* = 0.60). The regulation of many genes involved in vitamin biosynthesis and macro-molecule storage after microbiota loss, and the absence of a significant correlation between the loss of the microbiota and the hypoxia-induced signalling, support the hypothesis that the microbiota contributes metabolically to mosquito development.

**Investigation of folic acid supplementation and lipid accumulation during germ-free larval development.** To further investigate this role, we first focused on folate metabolism, as three key genes of this vitamin biosynthesis pathway were up-regulated in germ-free larvae (Fig. 6a). Since the *Ae. aegypti* genome lacks the genes that allow the conversion of 7,8-Dihydroneopterin to 7,8-Dihydrofolate (Fig. 6a, black dotted arrow[22]), we hypothesised that in the absence of a microbiota, the mosquito up-regulates folate biosynthesis genes as a response to a drop in folate levels. In support of this hypothesis, two genes annotated as folate transporters (*AAEL001687* and *AAEL001047*) were significantly up-regulated in all germ-free samples (Table S6), further suggesting that the mosquito responds to the lower folate levels in the absence of bacteria. To further explore the contribution of folate to larval development, we investigated the effect of folate supplementation on larval development. We observed that development success to adulthood was over 4-fold higher when providing folic acid to third-instar germ-free larvae upon *m*-DAP and D-Ala depletion, while development time was unaffected (Fig. 6b, development success to L4: *p* = 1.0; to pupae: *p* < 0.001; to adults: *p* < 0.001; Fig. S11, time to pupation: *p* = 0.26; L3: *p* = 0.57; L4: *p* = 0.25; metamorphosis: *p* = 0.34). When provided to first-instar axenic larvae, folic acid did not rescue larval development and larvae survived as first instar for 12–15 days before dying (Fig. S12). Considering that folic acid partly rescues development of third-instar larvae, we wondered whether this supplementation also had an effect on some genes detected in the RNASeq analysis as most differentially regulated, focusing on amino-acid and lipid storage and folate biosynthesis. Folic acid supplementation for 20 h from decolonisation did not rescue the expression of *hexamerin* genes or of genes implicated in lipid storage (Fig. S13). Moreover, the regulation of genes involved in folate biosynthesis was not directly related to folic acid supplementation (Fig. S13). We cannot exclude that other molecular intermediates participating in folate biosynthesis are sensed instead of folate in the transcriptional regulation of these genes.

We then explored the consequences of the second transcriptomic signal observed in germ-free larvae, the down-regulation of genes involved in amino-acid and lipid storage. To this aim, we quantified by fluorescence the lipid levels in the gut and in the pelt of germ-free larvae and in larvae carrying WT *E. coli*,

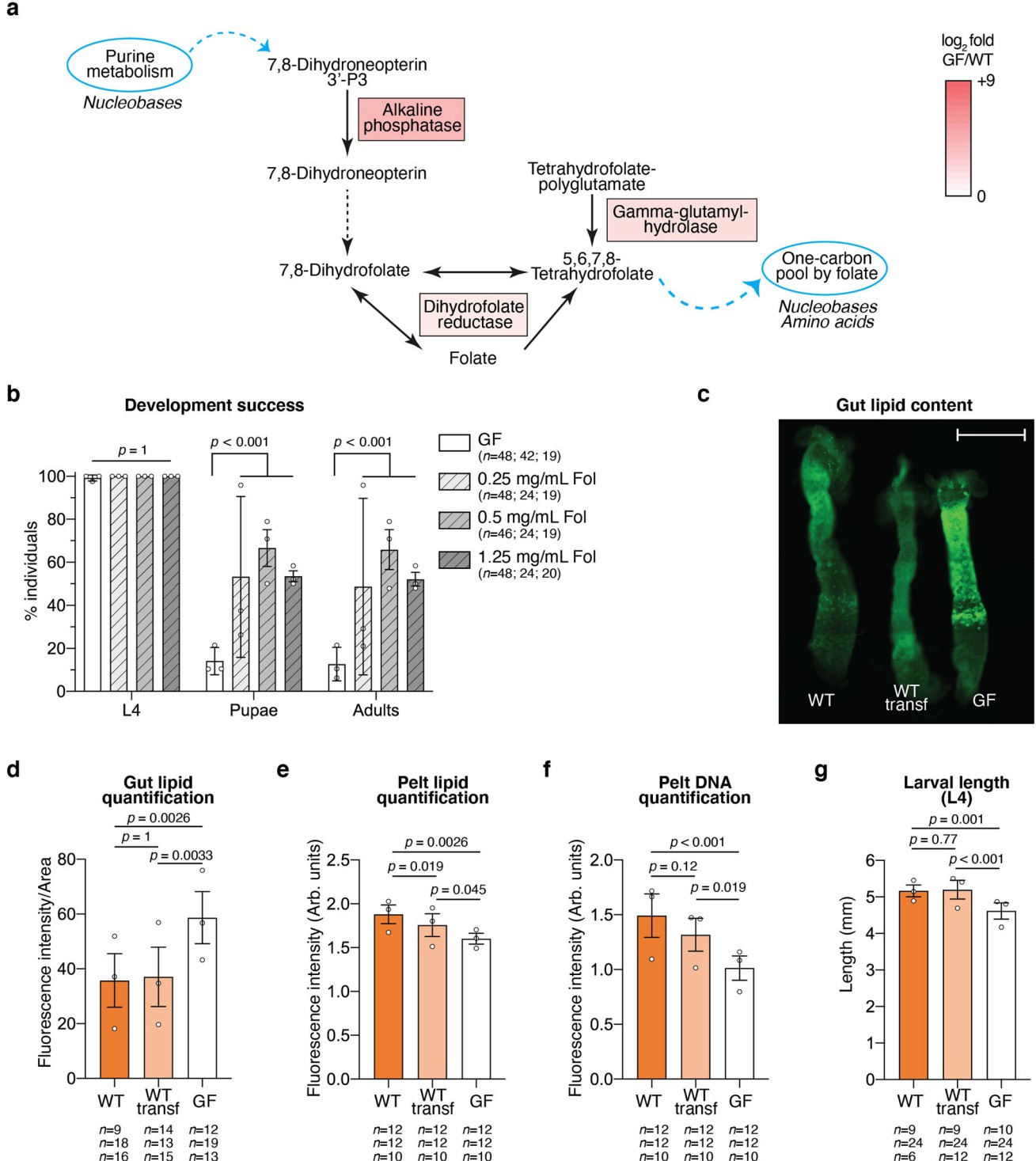

transferred or not transferred to sterile rearing water. We detected a 60% higher fluorescent signal in enterocytes of germ-free larvae compared to larvae carrying a microbiota (Fig. 6c, d, $p < 0.001$), while the amount of lipids in pelts, which are composed mostly of the cuticle and the fat body, was 15% lower in larvae that lost their microbiota (Fig. 6e, $p < 0.001$). The lower lipid levels detected in the pelt of germ-free larvae correlated with, and possibly led to, a slower growth of these larvae, as reflected by the lower DNA signal and the smaller larval size (Fig. 6f, $p < 0.001$; Fig. 6g, $p < 0.001$).

## Discussion

The mosquito microbiota impacts host physiology at the larval and adult stages via previously largely uncharacterised mechanisms. Our study led to the development of a method to produce fully developed germ-free adult mosquitoes. It allows investigation of the role of the microbiota in larvae and in adults independently and sheds light on the metabolic contribution of the microbiota during larval development.

The production of a relevant negative control for the study of the adult mosquito microbiota has long been a technical issue,

**Fig. 6 Investigation of folate supplementation and lipid metabolism in germ-free larvae. a** Representation of the mosquito gene products that participate in the folate biosynthesis pathway and that were up-regulated in germ-free larvae. Enzymes responsible for reactions between 7,8-Dihydroneopterin and 7,8-Dihydrofolate are not encoded by the *Ae. aegypti* genome (black dotted line). The up-regulation colour code is the same as in Fig. 5d and reflects the mean of $\log_2$ fold in the four sample types. Light blue circles represent metabolic pathways connected to folate biosynthesis through the indicated molecular products. Molecule types indicated below are involved in those pathways. **b** Proportion of larvae reaching the fourth instar (L4), the pupal stage and adulthood in germ-free conditions (GF, white) or in germ-free conditions with folic acid (Fol) supplementation at different concentrations (light grey: 0.25 mg/mL folic acid; mid grey: 0.5 mg/mL folic acid; dark grey: 1.25 mg/mL folic acid). **c** Exemplifying image of three guts belonging to fourth-instar larvae gnotobiotic for *E. coli* wild-type before (WT) and after the transferring to new rearing medium (WT transf) and germ-free (GF) stained with BODIPY 505/515. Scale bar: 1 mm. This image is representative of data from three independent replicates (see (**d**) for overall data and number of guts/replicate). **d, e** Quantification of BODIPY 505/515 fluorescence intensity in the gut (**d**, normalised to the gut area) and pelt (**e**, Arb. units – arbitrary units) of fourth-instar larvae gnotobiotic for *E. coli* wild-type before (WT, orange) and after the transferring to new rearing medium (WT transf, light orange) and germ-free (GF, white). **f** Quantification of DAPI fluorescence intensity in the pelt of fourth-instar larvae in the three gnotobiotic conditions. **g** Larval length measured at the second day of the fourth instar. In (**b, d–g**), data represent the mean ± SEM of three independent replicates. The exact number of individuals analysed per condition and replicate is indicated in each panel. Statistical significance was determined with generalised linear mixed models and least square means with Bonferroni correction. Exact *p* values are indicated in the figure. See Table S1 for detailed statistical information. Source Data are provided as a Source Data file.

which was either solved by allowing normal development and treating emerging mosquitoes with antibiotics or by producing stunted germ-free mosquitoes[7,9]. However, antibiotic treatments are known to reduce bacterial load and modify microbiota composition rather than produce germ-free mosquitoes[23], and toxic effects of antibiotics on the host cannot be ruled out[14]. Moreover, several pieces of evidence underline the requirement for standardised conditions during development due to carryover effects of the larval environment from larvae to adults. For instance, larval microbiota composition and larval diet affect host lifespan and vector competence, while crowding and starvation affect metabolic profiling in the adult fat body[10–12]. Here we provide a tool which allows the development of germ-free and colonised adults under comparable conditions, allowing further functional studies of the specific contribution of the microbiota in adults. We found that individuals reared with both AUX and WT *E. coli* showed similar life-history traits in terms of duration and success of development, larval and adult body-size, fecundity of the first gonotrophic cycle and lifespan.

Interestingly, in our system, fitness was slightly higher for gnotobiotic mosquitoes than for conventionally reared mosquitoes in terms of development and reproduction success. This may be due to the presence of bacteria with a low larvicidal activity in our colony, which could also explain the slight variation in sex ratio as females develop slightly slower than males and therefore may have prolonged contact with such pathogens. Alternatively, some individual larvae may not carry enough bacteria for successful larval development, which correlates with development success deficits after the transfer of colonised third-instar larvae. The lower propension to lay eggs and the smaller sized egg clutches of conventionally reared mosquitoes might be a consequence of the proliferation of the gut microbiota after the blood meal, which stimulates mosquito immunity and can have a negative effect on mosquito fitness[5,24,25]. The fact that the same negative effect on fecundity was not observed in *E. coli*-colonised mosquitoes could be due to a milder mosquito immune response to this bacterium.

The monitoring of bacterial CFUs in water during larval development suggests that bacterial intake peaks at the beginning of the fourth instar. This larger ingestion of bacteria might be explained by the bigger size of fourth-instar larvae and reflect the higher metabolic demand of the mosquito during the last larval stage prior to metamorphosis[26]. Bacterial counts significantly decreased in pupae reared on both *E. coli* strains, with more striking reductions observed in AUX pupae. This reduction in microbiota loads might be due to the feeding discontinuation before and during metamorphosis and to the induction of immune effectors, thought to avoid haemocoel invasion by the

microbiota during gut remodelling in other holometabolous insects[27]. This may also explain why no bacteria were recovered from approximately half of the examined adult mosquitoes originating from WT larvae. The lower bacterial loads found in fourth-instar AUX larvae and in the corresponding breeding water might be explained by the exhaustion of *m*-DAP and D-Ala by auxotrophic bacteria in the larval gut and their resulting inability to grow. This would be exacerbated by the interruption of feeding before metamorphosis and the resulting intestinal inaccessibility of *m*-DAP and D-Ala in pupae.

We were intrigued that larvae carrying either *E. coli* HS or a field-type microbiota, lost as much as 99% of their bacteria within 5 h after transfer into sterile water and that the microbiota went back to initial loads within a day. As similar results were obtained with larvae carrying *E. coli* and a native microbiota with minor kinetic differences, this drop is unlikely related to a default of adaptation of the *E. coli* HS strain to the mosquito gut. It may result from a dilution effect, as larvae were shown to completely excrete food and bacteria within a 30 min timeframe[8]. It is interesting to note that, although bacterial loads returned to initial values after 24 h, this microbiota perturbation was sufficient to significantly impact the development of *E. coli*-colonised larvae and, to a smaller extent, of larvae colonised by a conventional microbiota.

Our transcriptomic data points to gene up-regulation in the folate biosynthesis pathway and in some folate transporters during larval development in the absence of *E. coli* colonisation. Folate has a central role in development and represents a key enzymatic cofactor in the one-carbon metabolism, a series of reactions controlling the production of several important nucleic acid and amino-acid building blocks, such as purines, thymidine, formylated methionyl-tRNA, methionine, glycine and serine[28]. The genome of *Ae. aegypti* lacks several genes of the folate biosynthesis pathway, while the *E. coli* HS genome codes a complete biosynthetic pathway[29]. This suggests that the mosquito relies on its microbiota for the production of this vitamin. Accordingly, folate biosynthesis also appeared among the regulated pathways in adult *Anopheles* mosquitoes deprived of their microbiota[30], while *alkaline phosphatase*, one of the enzymes participating in folate biosynthesis, was up-regulated in germ-free *Ae. aegypti* adults[31]. Vitamins B have already been reported as a key contribution of the microbiota to development in other insects[32,33] and are essential nutrients for mice lacking a microbiota, together with vitamin K[34,35].

In mosquitoes, it appears that folate has a more central role in late larval development, as the folate pathway was not identified as microbiota-regulated in a previous transcriptomic study investigating the role of the microbiota on the mosquito early

larval development[35] and as folic acid alone is not sufficient to rescue larval development of first-instar axenic larvae. Older studies that aimed to define nutrients essential for the rearing of *Ae. aegypti* larvae in axenic conditions already pointed out the importance of folate in late larval development[16,36], but their results were not reproducible in more recent attempts[9]. In our system, only a small proportion of larvae usually completed their development to adulthood after losing their microbiota. Some egg batches originating from our colony led to higher development success of transferred larvae, suggesting that larval metabolism also depends on the metabolic status of the mother and consequently of the embryo. Germ-free larvae had a prolonged development, particularly marked for the fourth instar. However, once they entered the metamorphosis stage, almost all pupae completed their development, suggesting that the transition between larvae and pupae is a development checkpoint while pupal development is more even. The metamorphic onset is stimulated by a peak of ecdysone which is controlled by the size and metabolic status of the fourth-instar larva[36]. Folic acid supplementation increased the proportion of larvae passing this checkpoint and developing to adults, without restoring normal expression of genes involved in amino-acid and lipid storage in the conditions tested. We hypothesise that this higher proportion of individuals fulfilling the metamorphosis requirements is achieved by a combination of a prolonged fourth instar and a yet uncharacterised improvement in energy intake during the fourth instar.

Besides folate metabolism, our transcriptomic analysis showed that several lipid and amino-acid storage genes are expressed in a microbiota-dependent manner. We observed a lipid accumulation in gut enterocytes and a lipid deficit in the fat body of germ-free larvae, which was previously observed in axenic first instar *Ae. aegypti* larvae[21] and in germ-free *Drosophila* adults[37]. In insects, diet-derived triacylglycerols (TAG) are hydrolysed by TAG lipase via a two-step reaction in diacylglycerols (DAG), then monoacylglycerols, and free fatty acids (FFA) in the midgut lumen. FFA are then absorbed by the intestinal epithelium via an uncharacterised mechanism and stored in lipid droplets in enterocytes. FFA are then used as precursors for the synthesis of DAG and TAG that are loaded onto Lipophorin which transports them via the haemolymph to the fat body (reviewed in[38–40]). As two *lipophorins* encoding lipid transporters are down-regulated in germ-free larvae, the accumulation of lipids in the gut of these larvae might be explained by a reduced transport of lipids to the fat body. Bacteria also participate in lipid digestion by producing short-chain fatty acids (SCFA) as end-products of fermentation. A particular SCFA, acetate, is a key metabolite regulating lipid accumulation and metabolism in *Drosophila* enterocytes, as it controls development via IMD pathway-dependent signalling[37]. Our transcriptomic data did not identify any IMD pathway-regulated gene as microbiota-dependent, but the role of the microbiota for SCFA production is widely reported throughout the animal kingdom[41–43]. As folate and one-carbon metabolism are also correlated with lipid methylation and phosphatidylcholine synthesis through the methionine cycle[44,45], it is possible that our observations on lipid accumulation and on folate are somehow connected.

The most down-regulated gene family in the absence of a microbiota is Hexamerins, which are insect-specific proteins encompassing six subunits. They are amino-acid reserves stored in the fat body that are consumed during non-feeding stages and vitellogenesis[20]. They might also play a role in cuticle maturation and transport of riboflavin (vitamin B2) and hormones. Hexamerins accumulate during the end of larval development and represent as much as 60% of haemolymph proteins just before pupation[46]. In the mosquito, two sub-classes of Hexamerins exist,

namely Hexamerins-1 and -2, which are orthologues to *Drosophila* larval serum proteins-1 (CG2559, CG4178, CG6821) and -2 (CG6806), respectively, and are enriched in different amino acids. Repression of *hexamerins* might reflect a lower amino acid availability caused by a reduced peptidase activity or amino acid transport, two functions found to be down-regulated in the absence of bacteria in first instar axenic larvae[47]. However, in our system, peptidases were present in both up- and down-regulated genes (Table S3). This might reflect a reduced ability of larvae to digest diet-derived proteins in the absence of bacteria and, in parallel, an increased consumption of the proteins stored while still carrying a microbiota.

In conclusion, our study provides a new tool for the rigorous functional analysis of microbiota in mosquitoes, making it possible to distinguish between the contribution of the microbiota in larvae and adults. Moreover, the same protocol may be also applicable to produce adults in other insect species requiring gut bacteria for normal development. It allowed us to build a model for the metabolic contribution of the microbiota to larval development. We suggest that bacteria are required for folate biosynthesis, which supports the later stages of development, and for energy storage in the form of proteins and lipids, which promotes larval growth.

## Methods

**Ethics statement**. No protected animal was used for the experiments described in this paper. For the maintenance of the mosquito colony, mosquitoes were blood-fed on mice. Protocols have been validated by the French Direction générale de la recherche et de l'innovation, ethical board # 089, under the agreement # 973021.

**Bacteria and mosquitoes**. *E. coli* HS (wild-type, WT) and *E. coli* HA416 (auxotroph, AUX[17,48]) were grown in lysogeny broth (LB). *E. coli* HA416 cultures were supplemented with 50 μg/mL kanamycin, 50 μg/mL *meso*-diaminopimelic acid (*m*-DAP) and 200 μg/mL D-alanine (D-Ala). For all experiments, *E. coli* cultures were inoculated from single fresh colonies in liquid LB with appropriate supplementation and incubated at 30 °C, shaking at 200 rpm for 16 h. Growth curves of AUX *E. coli* with optimal amino-acid concentrations (50 μg/mL *m*-DAP and 200 μg/mL D-Ala) and concentrations compatible with first-instar larvae (12.5 μg/mL *m*-DAP and 50 μg/mL D-Ala) were obtained by inoculating 2 mL of LB supplemented with the two amino-acid concentrations with the same diluted bacterial culture in a 24-well plate. The optical density at 600 nm ($OD_{600}$) was monitored every 15 min using a FLUOstar plate reader (BMG Labtech). The experiment was performed two times.

The *Ae. aegypti* New Orleans colony was maintained under standard insectary conditions at 28–30 °C on a 12:12 h light/dark cycle. Gnotobiotic mosquitoes were maintained in a climatic chamber at 80% relative humidity on a 12:12 h light/dark 30 °C/25 °C cycle.

**Generation of germ-free (axenic) larvae**. Inside a microbiological safety cabinet, eggs were scraped off the paper they were laid on and transferred onto a disposable vacuum filtration system. Eggs were surface sterilised via incubation in 70% ethanol for 5 min, then in 1% bleach for 5 min and finally in 70% ethanol for 5 min. Bleach solution was extemporaneously prepared from a tablet. After each incubation step, a vacuum was applied to the filtration unit for fast replacement of the sterilising liquid. After rinsing three times with sterile water, the eggs were resuspended in ~20 mL of sterile water and transferred to a sterile 25 mL cell-culture flask with a vented cap. Approximately 20–30 eggs were inoculated in 3 mL of liquid LB and incubated for 48 h at 30 °C shaking at 200 rpm to confirm sterility. Flasks were kept standing in a climatic chamber at 80% relative humidity on a 12:12 h light/dark 30 °C/25 °C cycle for eggs to hatch overnight.

**Generation of gnotobiotic larvae and axenic adults *via* reversible colonisation**. Depending on the experiment type, two rearing methods were used: for experiments aiming to analyse adult mosquitoes, a batch rearing in cell-culture flasks (10–15 larvae per flask) was used; for experiments requiring the follow-up of single larvae, larvae were placed in single wells of sterile 24-well plates[49].

A 16 h culture of each *E. coli* strain was pelleted and cells were resuspended in 5 times the initial culture volume of sterile deionised water to reach a concentration of 2–5*10^8 CFU/mL. For the generation of larvae gnotobiotic for AUX *E. coli*, *m*-DAP (12.5 μg/mL) and D-Ala (50 μg/mL) were also added to the bacterial suspension. For batch rearing, 20 mL of bacterial suspension and 750 μL of autoclaved suspension of TetraMin Baby fish food (50 mg/mL in water; composition: 46% proteins, 11% fats, 2% cellulose, 17,310 IU/kg vitamin A; 1080 IU/kg vitamin D₃, 96 mg/kg manganese, 57 mg/kg zinc, 37 mg/kg iron,

0.7 mg/kg cobalt) were added to each flask. For rearing of individualised larvae, 2 mL of bacteria and 50 μL of fish food suspension were added to each well. Larvae were kept in the climatic chamber until pupation.

Pupae were rinsed once in sterile water and then transferred to 15 mL tubes placed in autoclaved polypropylene cups (Eco2 NV) for emergence. A smaller tube containing a rod of sterile cotton dipped into a 10% sterile sucrose solution was placed in each cup. Tubes containing pupae and sugar were held in place using specifically designed 3D printed components (designs available upon request). See Fig. S14 for detailed information on the preparation of sterile cups.

**Blood feeding of mosquitoes in sterile conditions.** Gnotobiotic pupae were rinsed in sterile water and transferred into a sterile cup for adult rearing without providing any sucrose solution. At the fifth day after emergence, mosquitoes were offered a sterile blood meal consisting of 50% human red blood cells (Etablissement Français du Sang de Guadeloupe-Guyane) and 50% fetal bovine serum (Thermo Fisher). The blood meal was provided to mosquitoes in an autoclaved Hemotek reservoir (Hemotek) replacing the collagen membrane with a Parafilm sheet (Bemis) sterilised by a 10 min $H_2O_2$ treatment. Mosquito cups were kept inside an MSC for the whole blood-meal duration.

**Duration of development.** All measurements for the investigation of the developmental dynamics were performed on WT and AUX *E. coli*-carrying gnotobiotic individuals and from conventionally reared individuals obtained from non-sterilised eggs and reared in non-sterile conditions (using tap water as conventionally used for our mosquito colony). Three independent mosquito batches were examined for each type of analysis.

Gnotobiotic or conventionally reared larvae were placed in single wells of 24-well plates to monitor the duration of the different developmental stages in single individuals and to determine the developmental success of each rearing condition, measured as the percentage of larvae reaching the adult stage. Plates were inspected three times per day and the moment at which larvae moulted was noted. The sex of the adult mosquito or the time of death were also recorded for each individual. The experiment was stopped after 15 days, generally 5 days after the last emergence. Individuals stacked at the larval stage were marked as non-developed. For each replicate, at least 45 individuals per condition were analysed.

**Reversible colonisation dynamics.** To validate reversible colonisation in adults, mosquitoes emerging from AUX larvae were collected 0–2 days after emergence, surface sterilised for 3–5 min in 70% ethanol, rinsed three times in sterile PBS and homogenised individually in 100 μL of sterile LB. To quantify the number of bacterial CFUs present in each individual, serial dilutions of the homogenates were plated on LB agar plates supplemented with 50 μg/mL *m*-DAP and 200 μg/mL D-Ala. Fig. S3 shows data from three independent replicates based on 34, 123 and 321 individuals.

To investigate colonisation dynamics of AUX *E. coli*, the bacterial loads in third- and fourth-instar larvae and pupae were investigated and compared to those of individuals carrying WT *E. coli*. Larvae and pupae were surface sterilised for 15–30 s in 70% ethanol, rinsed in sterile water and homogenised individually in 100 μL of sterile LB. Serial dilutions of the homogenates were plated on LB agar plates supplemented with 50 μg/mL *m*-DAP and 200 μg/mL D-Ala and the number of bacterial cells present in each individual was determined. Three replicates of at least 8 individuals per replicate were analysed at each time-point.

**Detection of bacterial DNA in midguts and whole mosquitoes.** To detect and quantify the presence of bacterial DNA on adults, mosquitoes were collected 7–9 days after emergence, surface sterilised for 3 min in 70% ethanol and rinsed three times in sterile PBS. DNA was extracted from pools of 10 midguts of sugar-fed mosquitoes, pools of 4–10 midguts of blood-fed mosquitoes and pools of 10–16 sugar-fed whole mosquitoes. Dissected tissues or whole mosquitoes were placed in 2 mL screw-cap tubes containing 0.5 mm glass beads and homogenised using a bead beater (Precellys Evolution, Bertin) for 2 × 1 min at 9000 rpm in 250 μL (midguts) or 600 μL (whole mosquitoes) easyMAG Lysis Buffer (Biomérieux). Three blank samples were extracted in parallel and used as negative controls. After 10 min incubation DNA was purified using a Nuclisens easyMAG apparatus (Biomérieux). The presence of bacterial DNA was detected and quantified via amplification of the *16S* rRNA bacterial gene and the ribosomal mosquito gene *S17* via qPCR using the SYBR qPCR Premix Ex Taq (Takara) in a LightCycler 480 (Roche). Primer sequences are listed in Supplementary Table S7. Mosquitoes were considered germ-free when their *16S* Ct were higher than those of negative controls or when any amplicon was detected. Three independent replicates were analysed.

**Larval and wing length measurements.** For larval length measurements, larvae were collected at the fifth day from hatching. Larvae were fixed in PBS with 4% paraformaldehyde for 20 min and placed on a microscope slide. Imaging was performed on an Euromex dissection microscope equipped with a camera. For each larva, the distance between the anterior border of the head and the posterior border of the last abdominal segment excluding the syphon[49] was measured using the ImageFocus software (version 4, Euromex). Four independent mosquito batches

were examined and, for each replicate, at least 27 individuals per condition were analysed.

For wing length measurements, adult mosquitoes were collected 2 days post emergence and kept at −20 °C until analysis. The measurements were conducted on the left wing of both male and female mosquitoes. Wings from several individuals were dissected and placed on a microscope slide. The distance between the alular notch and the radius 3 vein was measured for each wing using a dissection microscope equipped with a camera[49]. Four independent mosquito batches were examined and, for each replicate, 7–20 individuals per condition were analysed.

**Fertility and fecundity.** Mosquitoes were blood-fed in sterile conditions. After the blood meal, mosquitoes were anesthetised with $CO_2$ and individually placed in autoclaved polypropylene *Drosophila* vials (Flystuff) equipped with a micro-centrifuge tube for mosquito allocation, a piece of Whatman™ paper for egg recovery and ~1 mL sterile water. After 4 days, the number of mosquitoes laying eggs and the number of eggs laid per mosquito were recorded. To check for mosquito contamination, 3 mL of LB (or LB supplemented with *m*-DAP and D-Ala) were added to the empty *Drosophila* vials and incubated shaking at 200 rpm for 48 h at 30 °C. Once dried, paper sheets on which eggs were laid were stored in non-sterile conditions. They were placed in water in individual containers to determine viable egg-clutches. Three independent replicates were performed using 12–25 females per condition.

**Lifespan.** Gnotobiotic pupae were rinsed in sterile water and aseptically transferred into a 96-well plate for the sex sorting based on pupal size. For each sex, 19–28 pupae were transferred into sterile cups for adult rearing. The number of dead mosquitoes was recorded daily. Three independent replicates were performed.

**Analysis of bacterial loss dynamics and decolonisation.** For the analysis of bacterial loss dynamics, at $T_0$, third-instar larvae carrying the WT or the AUX *E. coli* were washed in sterile water and individually transferred to new 24-well plates containing 2 mL of sterile water and 50 μL of sterile fish food. Non-transferred larvae were reared in parallel for 2 h. After 2, 5, 12 and 20 h from the transfer, at least 6 larvae per condition were surface sterilised for 15–30 s in 70% ethanol, rinsed in sterile water and homogenised individually in 100 μL of sterile LB. Serial dilutions of the homogenates were plated on LB agar plates supplemented with 50 μg/mL *m*-DAP and 200 μg/mL D-Ala and the number of bacterial cells present in each individual was determined. Three replicates were analysed at each time-point.

To validate decolonisation at different larval stages, gnotobiotic larvae individually reared in 24-well plates were transferred every 24 h into new plates containing 2 mL of sterile water and 50 μL of sterile fish food. Bacterial counts were measured in 6 larvae per condition after 16 h from transfer as described above. Three replicates were analysed at each time-point.

To analyse the bacterial loss in larvae colonised by a native microbiota, water was collected from *Ae. aegypti* breeding sites in Cayenne (French Guiana) and added to axenic first-instar larvae individually placed in 24-well plates. After 48 and 72 h, half of larvae were washed in sterile water and individually transferred to new 24-well plates containing 2 mL of sterile water and 50 μL of sterile fish food. Non-transferred larvae were reared in parallel. After 2, 5 and 24 h from the transfer, bacterial counts were measured in 5–6 larvae per condition. Two and three replicates were analysed for the 48 and 72 h time-points, respectively.

**Analysis of development in transferred larvae.** Larvae reared with WT and AUX *E. coli* and larvae colonised by a native microbiota were obtained as described above and transferred to new rearing medium at the selected time-point. Non-transferred larvae were reared in parallel. To investigate the effect of folate on development success, 0.25, 0.5 or 1.25 mg/mL folic acid were added to germ-free larvae. Plates were inspected daily and the developmental stage of each individual was noted. At least three replicates were performed using independent mosquito batches. With some egg batches we observed that all larvae completed their development despite their rearing condition. We discarded those data and we considered valid and experiment when WT *E. coli* gnotobiotic larvae transferred to new rearing medium had a 40–60% developmental success. For each replicate, 12–58 larvae were analysed.

**Experimental setup and sample collection for transcriptomics on colonised vs germ-free larvae.** Gnotobiotic larvae for both bacterial strains were obtained as described above, reared individually in 24-well plates and kept in a climatic chamber at 80% RH with 12:12 h light/dark 30 °C/25 °C cycle. AUX larvae were supplemented with *m*-DAP (12.5 μg/mL) and D-Ala (50 μg/mL). To ensure synchronicity, larvae moulting to third instar between 48 h and $T_0 = 53$ h after addition of bacteria were selected. At $T_0$, AUX larvae were washed in sterile water and individually transferred to new 24-well plates containing 2 mL of sterile water and 50 μL of sterile fish food. Twelve hours later ($T_{12}$), 120 larvae per condition were collected for sampling and another 120 AUX larvae were transferred to new 24-well plates containing 2 mL of sterile water and 50 μL of sterile fish food to further reduce bacterial colonisation. Another 8 h later ($T_{20}$), 120 larvae per condition were collected for sampling.

At both $T_{12}$ and $T_{20}$, for each condition, 60 larval guts (dissected on ice) and 60 whole larvae were snap frozen in prechilled 2 mL screw-cap tubes containing 0.5 mm glass beads and kept in a cooling block below −40 °C. A gut sample included gastric caeca, anterior and posterior midgut, while hindgut and Malpighian tubules were excluded. Practically, $T_0$, $T_{12}$ and $T_{20}$ of each condition was 1 h apart to match our dissection speed (60 individuals/h) and the order of processing was inverted between replicates. Three replicates were performed using independent mosquito batches.

**RNA extraction, library preparation, RNA sequencing and data analysis**. After collection, 1 mL of TRIzol (Thermo Fisher Scientific) was added to all samples. Samples were homogenised using a bead beater (Precellys Evolution, Bertin) for $2 \times 1$ min at 9000 rpm and stored at −80 °C until RNA extraction. Total RNA was extracted using TRIzol according to the manufacturer's instructions. After RNA extraction, samples were treated for 30 min at 37 °C with 10 U of DNase I (Thermo Fisher Scientific) and purified using the MagJET RNA Kit (Thermo Fisher Scientific) in a KingFisher Duo Prime system (Thermo Fisher Scientific). Samples were mixed with RNA stable (Biomatrica), vacuum dried using a SpeedVac and shipped for library preparation and sequencing.

RNA quality evaluation, library preparation and sequencing were performed by Biofidal (Vaulx-en-Velin, France - http://www.biofidal-lab.com). RNA integrity was evaluated on a BioAnalyzer (Agilent Technologies) using an RNA 6000 Pico RNA chip. Single-end libraries (75 bp) with poly(A) selection were constructed using the Universal Plus mRNA-Seq Library Preparation with NuQuant kit (Nugen Tecan Genomics) from 1 µg of total RNA. After quality check, libraries were pooled in equimolar concentrations and sequenced on a NextSeq 500 high-output flow cell (Illumina). The resulting raw reads were de-multiplexed and adaptor sequences were trimmed by the sequencing facility. FASTQ sequences were trimmed for their quality using Trimmomatic[50] (version 0.39, minimum quality score: 15, minimal length: 40) and their quality was evaluated using FASTQC[51] (version 0.11.9). After quality trimming, an average of 15 million reads (min 8.4 million, max 25 million) were obtained per sample. Reads were mapped on the *Ae. aegypti* genome (assembly AeaegL5, geneset AeaegL5.2) using HISAT2[52] (version 2.1.0) with a mean alignment rate of 94%. Read counts were obtained with featureCounts[53] (version 1.6.4) and differential expression analysis was conducted in R using the DESeq2[54] package (version 1.24.0). Only the genes with an adjusted $p$-value < 0.001 and a $\log_2$ fold change < −1.5 or >1.5 were subjected to the Gene Ontology analysis with g:Profiler[55].

To quantify bacterial loss on samples subjected to RNA sequencing, RNA was retro-transcribed using the PrimeScript RT-PCR Kit (Takara) and analysed via qPCR using the SYBR qPCR Premix Ex Taq (Takara) in a Ligth Cycler 480 (Roche). Bacterial load was quantified via amplification of the *16S* rRNA bacterial gene and normalised to the ribosomal mosquito gene *S17* using the following formula: $R = E_{\text{target}}^{-\text{Ct target}} / E_{\text{reference}}^{-\text{Ct reference}}$, with $E$ representing primer efficiency. Statistical analyses were performed on $\log_2$ R values. Primer sequences are listed in Table S7.

**Expression analysis after folic acid supplementation**. Third-instar colonised and germ-free larvae were obtained as described above. At the time of transfer ($T_0$) 1.25 mg/mL folic acid was added to half of the larvae. At $T_{20}$, 18–24 larvae per condition were sampled and immediately stored at −80 °C. Three replicates were performed using independent mosquito batches, except for larvae reared with WT *E. coli* and transferred to new rearing medium which were sampled from two replicates. RNA was extracted as described above, retro-transcribed using the PrimeScript RT-PCR Kit (Takara) and analysed via qPCR using the SYBR qPCR Premix Ex Taq (Takara) in a 7300 Real-Time PCR System (Applied Biosystems). Data analysis was conducted as described above using the ribosomal mosquito gene *S17* as reference gene. Primer sequences are listed in Table S7.

**Gut hypoxia measurements**. Third-instar colonised and germ-free larvae were obtained as described above. At least 9 larvae per condition were collected at the 12 h time-point, washed in sterile water and incubated in the dark for 30 min in sterile PBS supplemented with 5 µM Image-iT Red Hypoxia Reagent (Thermo Fisher Scientific) and fish food. In parallel, two larvae per condition were processed without the dye as a negative control. Larvae were frozen for 3 min at −80 °C and placed in a single slide. Images were acquired with an EVOS FL Auto system (Thermo Fisher Scientific) equipped with a Texas Red filter using a 10X objective. The multiple images were stitched automatically using the EVOS FL Auto Software to reconstitute a single image including all samples. Fluorescence intensities and area surfaces of single guts were measured using Fiji[56] (version 2.0.0). Three replicates were performed using independent mosquito batches.

**Lipid content and length measurements**. Third-instar colonised and germ-free larvae were obtained as described above. Larvae were sampled 44 h after the transfer, when WT larvae were at the second day of the fourth instar. Guts and pelts (composed by the cuticle and the fat body) were dissected, placed in PBS with 4% paraformaldehyde and fixed overnight at 4 °C.

After three 5 min washes in PBS, guts were stained in the dark for 20 min with DAPI (Thermo Fisher Scientific, final concentration 0.1 µg/mL) and for 20 min with BODIPY 505/515 (Thermo Fisher Scientific, final concentration 1 µM), rinsed three times in PBS for 5 min and imaged. Imaging and analysis were performed as

described above, using DAPI and GFP filters. A different protocol was used to quantify lipids in the pelts to avoid the loss of fat body parts during staining and rinsing steps. After fixation, pelts were washed three times in PBS for 5 min and placed in individual tubes with 75 µL of DAPI (0.1 µg/mL) and 75 µL of BODIPY 505/515 (1 µM). Pelts were homogenised, incubated in the dark for 20 min and transferred to a 96-well plate. Fluorescence intensities were measured using a FLUOstar plate reader (BMG Labtech) equipped with DAPI and GFP filters, using a solution of the two dyes as blank. At least 9 larvae per condition were analysed in three independent replicates.

Larval lengths were measured as described above on larvae obtained with the same protocol and collected at the 44 h time-point. Three independent replicates were performed using 6–24 individuals per condition.

**Statistical analyses**. Data were collected and organised in Excel for Mac (version 16.16.27). Graphs were created with GraphPad Prism (version 8.4.3). Statistical analyses were performed with generalised linear mixed models (GLMM) using the lme4 package in R (version 3.6.0) and setting the replicate as a random effect. For categorical data (development success, sex ratio, egg laying success, hatching success) an ANOVA was performed on a logistic regression (glmer). For quantitative data (CFU counts, duration of development, larval and wing length measurements, clutch size, gene expression, hypoxia, lipid and DNA quantifications) an ANOVA was performed on a linear regression (lmer). For survival analyses, a Cox proportional hazards model was performed. Detailed statistical information and number of individuals analysed per replicate can be found in Table S1. Variations are calculated as the difference between treatments normalised to the control.

**Reporting summary**. Further information on research design is available in the Nature Research Reporting Summary linked to this article.

## Data availability
The raw RNA sequencing data underlying Fig. 5 are available in the Sequencing Read Archive (SRA) under the Bioproject ID PRJNA687261. Mosquito cartoons are available under a CC-BY-NC-SA 4.0 license on our laboratory website (https://microbiota-insect-vectors.group/en/diy). Materials and additional data are available from corresponding authors upon request. Source data are provided with this paper.

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

## Acknowledgements

We thank Jean-Géraud Issaly for egg production; Charles Bobin and Verena Kircher for preliminary work on some protocols; Chloé Gapp for technical help; Albane Imbert and the Institut Pasteur FabLab for 3D prints design; Emma Wise for English proofreading of the manuscript. BODIPY is a kind gift from Philippe Sansonetti's laboratory. Data analysis was facilitated by a Bioinformatic Course funded by EU H2020 INFRAVEC2 (Project no. 731060) that OR attended. This work is funded by the French Government's Investissement d'Avenir program, Laboratoire d'Excellence "Integrative Biology of Emerging Infectious Diseases" (grant no. ANR-10-LABX-62-IBEID) and by ANR JCJC MosMi to MG (grant no. ANR-18-CE15-0007). S.H. received funding from the Swiss National Science Foundation (www.snf.ch; grant 169791).

## Author contributions

M.G. conceived the idea of applying reversible colonisation to mosquitoes. S.H. provided *E. coli* strains and expertise. M.G. and O.R. designed experiments. O.R., J.C.S and M.G. performed experiments. O.R. analysed data and prepared the figures. M.G. and O.R. co-wrote the manuscript. O.R., M.G. and S.H. revised the manuscript. M.G. acquired funding for this project. All authors accepted the final version of the manuscript.

## Competing interests

The authors declare no competing interests.
