## [Peer Review File · Nature Communications]

Reviewers' Comments:

Reviewer #1:

Remarks to the Author:

In this manuscript, Romoli et al., developed a rigorous protocol to generate fully developed germ-free adult *Aedes aegypti* mosquitoes. The protocol is based on a reversible colonization of bacteria genetically modified to allow complete decolonization at the end of the larval development without affecting adult size. In addition, the authors conducted multiple experiments including transcriptome analysis, diet supplementation, measurement of oxygen levels, and lipid analysis to evaluate the metabolic contribution of the microbiota during mosquito larval development. The authors concluded that bacteria contribute to mosquito larval development via folate biosynthesis and energy storage, and that folate supports larval development in the absence of a microbiota. The findings are sound.

I have listed a few specific comments below:

1-Line 170: In the text, it is not clear how the transcriptome analysis was validated.

2-Lines 193-195: "As many genes of these pathways are missing in the genome of *Ae. aegypti*, this suggests that, as reported in other insects, the mosquito microbiota participates in vitamin B".

References are missing here.

3-Lines 197-199. "Hexamerins are proteins reported to participate in amino-acid storage during the end of larval development and metamorphosis." References are also missing here.

4-Lines 321-322: "The metamorphic onset is stimulated by a peak of ecdysone which is controlled by the size and metabolic status of the larva". References are also missing here.

5- Line 341: Review written by Gondim et al. (*Insect Biochem Mol Biol*. 2018, 101:108-123) should be included in the discussion section.

6-Line 414: What is the composition of the fish food used for rearing mosquito larvae?

7-Lines 560-570: How was the normalization of data conducted?

8-Lines 627-718. References Section contains several mistakes, typos or omissions. Some names of the journals are abbreviated, others not. Some titles of the articles are capitalized others not. Please take a close look at the Author's guides for Nature Communications.

9-Lines 632, 638, 641, 642, 643, 648, 654, 659, 662, 663, 669,672, 681, 687, 690, 695, 701: italics in scientific names are missing here.

10-Line 641: please capitalize the first letter (A) of *Aedes* and use italics for *Aedes aegypti*.

11-Line 643. Do not capitalize a of *aegypti*

12-Line 685: Why does this reference have an asterisk?

13-Line 699: The name of the journal is missing.

14-Line 706: The name of the journal is missing.

15-Line 760: A or Ae to abbreviate *Aedes*, please be consistent

16-Supplementary material:

a) Figure S1-S10: Why are the number and the title of each figure repeated twice in each figure?

b) Figure S3: How were the data normalized? What statistical analysis was performed?

c) Figure S4-S6, S8: What statistical analysis was performed?

d) Figure S9: How was the gene expression monitored? How were the data normalized? In the legend of this figure, you should also mention the Table inserted in the figure.

e) Figure S10: Please include scale bars, and consider labeling important items.

f) Table 1S-4S. Please, provide more details. Such as how were the data obtained?

g) Table S5: Legend of this table is incomplete.

h) In some cases, it is indicated qPCR, and in other cases it is indicated RT-PCR. Please be consistent.

17-Supplementary references: See comments above. Please be consistent with the name of journals, titles of the articles, etc. Please use italics for scientific names of organisms.

Reviewer #2:

Remarks to the Author:

In the manuscript "Bacteria contribute to mosquito development via folate biosynthesis and energy storage", Romoli et al. apply a decolonization method previously characterized in mice to mosquitos in order to generate Germ-Free (GF) adults insects. They associate GF eggs with *E. coli* HA416, which is auxotroph for two amino acids required for peptidoglycan biosynthesis. They supplement the mosquito diet with these amino acids, allowing *E. coli* colonization of larvae and thus development. When the mosquitoes reach the adult stage, or when 3rd instar larvae are transferred to a medium deprived of these amino acids, *E. coli* cannot grow anymore and the mosquitoes are rendered GF. The authors use this method to compare the transcriptomes of GF larvae and mono-associated larvae. They find differences in expression of genes involved in folate and thiamine metabolism, lipid utilization and amino acids storage. In line with previous studies, they show that folate is essential to the development of GF larvae, and that GF larvae have impaired lipid utilization.

The method presented here, which was introduced and characterized in previous works in mice, is well described and well adapted for the rapid generation of GF adults mosquitoes. This method will be useful for gnotobiotic studies on adult mosquitoes as it is probably better than the use of antibiotics. However, the main messages of the paper are not really new in the field of insect-bacteria interactions:

- The lack of GF mosquitoes has long been an issue, but as the authors cited, it was recently made possible by the work of Correa et al. where GF larvae are reared in sterile medium supplemented with an agar plug containing nutrients. In those conditions, mosquitos present a developmental delay and reduced adult size. The authors seem to consider this problematic, however stunting in GF animals has been observed in other insect models such as *Drosophila*, which suggests that animals need microbes for optimal development.
- The RNAseq results provide interesting descriptive data, but they do not lead the authors to any novel mechanism because current literature already shows the microbial contribution to folate needs of animals (Blatch et al., 2010, Piper et al., 2013, Engevik et al., 2020).
- Finally, the influence of bacteria on lipid utilization (Valzania et al., 2018) was already known.

Major issues:

- The decolonisation method is interesting and well executed in mosquito, however it uses a gut-adapted human commensal strain. Conclusions thus cannot be extrapolated to the mosquito microbiota.
- The title of the article does not clearly reflect the content because the main objective of the study is the adaptation and testing of the reversible colonization method.
- The claim in the title that "Bacteria contribute to mosquito development via folate biosynthesis and energy storage" is not well supported by the data. Gnotobiotic experiments such as colonization with WT members of the mosquito microbiota and their folate biosynthetic genes knock-out counterparts will be needed. Quantifying folate production by commensal bacteria would also help to support this claim.
- The decolonization method only produces GF 3rd instar larvae and adults. Conclusions on the impact of bacteria on development are very restricted since they are limited to the 3rd instar larvae which is a developmental stage of "preparation" to metamorphosis. Is it possible to decolonize earlier? If microbiota is important for nutrition it may also impact earlier larval stages.
- Microbial impact on host physiology is dependent on active bacterial metabolism. The authors showed colonization of mosquito larvae by the bacteria tested but are bacteria actually growing and replicating in the mosquito diet upon amino acid supplementation? Or are they just persisting, alive, without doubling? It would be interesting to follow bacterial growth through time in the mosquito diet and the changes in bacterial counts upon amino acids supplementation in presence or absence of mosquito larvae.
- The authors should test whether reversible colonization allows to re-associate GF mosquito larvae with other bacteria: if there are some remaining *E. coli* AUX in the gut or in the water, they may take

advantage of amino acids produced by these other bacteria to grow again. The authors should perform re-association experiments at different time points after transfer, and test for the presence of *E. coli* AUX in the gut and in the water.

Minor issues:

- L46: "In the past, several studies proposed rearing protocols of germ-free larvae, which could not be reproduced in recent experiments⁸⁻¹⁰" the authors should not cite the work of Correa et al. here. It suggests that this work could not be reproduced. If that is what they actually mean, they should state it and provide references.
- L230: "These observations suggest that bacteria are essential throughout all larval development" Is it possible to decolonize at an earlier stage?
- L301: "the *E. coli* genome codes a complete biosynthetic pathway except for the gamma-glutamyl-hydrolase" the authors cite KEGG, which shows the pathway for a reference strain of *E. coli*, but strains may have different biosynthesis abilities. What about the strain used in this study and the actual mosquito microbiota strains?
- L302 and L314: "Our transcriptomic and diet supplementation data now provide strong evidence that bacteria and mosquito cooperate for folate biosynthesis" Can this *E. coli* strain produce THF-polyglutamate? If yes, can the authors test a knock-out mutant for THF-polyglutamate production? Can the authors provide THF-polyglutamate to the decolonized larvae and verify whether it has the same effect as folate? All these experiments are needed to support this claim. Otherwise I would recommend being more cautious in the phrasing.
- L360: Recent work shows bacterial influence on the host's expression of folate transporters (Engevik et al., 2020) did the authors find differences in regulation of folate transporters?
- The authors show that upon reversible colonization, there is no more culturable *E. coli* AUX at the adult stage (Fig 1D). However, they do find copies of 16S DNA (Fig S3) at D1 after eclosion. Are these copies from dead *E. coli*? Might some *E. coli* survive the lack of amino acids by entering a viable but non-culturable stage? It would be interesting to know whether these copies persist later in the adult stage.
- Fig 4A: it is unclear to me whether the extra lipids in GF guts are in the lumen of the gut, suggesting defects in lipid uptake, or in the enterocytes, suggesting defects in lipid utilization (as it was observed in *Drosophila*, Kamareddine et al 2018).
- The authors focused their findings on the bacterial contribution to mosquito development through the regulation of host's folate metabolism, however the discussion on the role of folate in development and metabolism is scarce.

Reviewer #3:

Remarks to the Author:

In this manuscript, Romoli et. al. describe a novel technique to decolonize *Aedes aegypti* of their microbiome. They use a strain of *E. coli* that is auxotrophic for certain metabolites, when these are removed from the larval rearing environment the bacteria stop growing and over some time the larvae become axenic.

Developing such a system is a major step forward in the research on mosquito / insect microbiomes. I think that this methodology is well-thought out and the experiments to demonstrate how it works are well thought out.

Major points:

1. To me it is quite surprising that the authors develop a strategy to make axenic mosquitoes (that are not compromised in the way other authors have achieved this in the past) but then carry out little to no investigations on the adult stage of the mosquito. Surely this is the most relevant from many

standpoints but especially from the standpoint of disease transmission. Would have been interesting to see if axenic mosquitoes generated this way had any effects on lifespan / fecundity and also to have examined gene expression in the adult stage.

2. The authors observe that mosquito's shed their bacteria when transferred to sterile water. It wasn't clear to me if this effect is E.coli specific or if the natural microbiome is also lost under these conditions. I wonder if the natural microbiome have a better ability to be retained under these conditions.

3. Similar to the point above, authors state on line 235, "A 60% reduction in developmental success was also observed among wild-type E.coli carrying larvae" - is this effect E.coli specific or would it also happen with the natural microbiome?

4. Its not clear to me if the authors investigated whether Folic acid was sufficient for larval development across all instars in the absence of bacteria.

5. Line 257-267 This section isn't well developed. It seems like a bit preliminary. Have the authors investigated the literature to determine if there is any potential link between vitamin biosynthesis and lipid transport? What about hemolymph lipid content? Is there any theory as to why without bacteria DAG and TAG cannot be generated in the enterocytes? The larval accumulation has been observed before, I wonder if there is a similar effect in adults, which has not been investigated or at least not investigated with a system with minimal limitations (reversible colonization).

6. The structure of the results could be improved. For example the "gut lipid" and "hypoxia" sections are a bit lost within other results sections. While important to introduce the methods, we aren't entirely sure what the key finding of the paper is, axenic technique or folic acid?

Minor points:

1. Line 176 Important could be replaced with significant or distinct

2. Line 320 "cannot" is probably a bit strong in this context.

3. Lines 205-223 The authors spend a lot of time refuting the 'hypoxia' theory, while interesting I feel that this could be refined/ shortened.

REVIEWER COMMENTS

Reviewer #1 (Remarks to the Author):

In this manuscript, Romoli et al., developed a rigorous protocol to generate fully developed germ-free adult *Aedes aegypti* mosquitoes. The protocol is based on a reversible colonization of bacteria genetically modified to allow complete decolonization at the end of the larval development without affecting adult size. In addition, the authors conducted multiple experiments including transcriptome analysis, diet supplementation, measurement of oxygen levels, and lipid analysis to evaluate the metabolic contribution of the microbiota during mosquito larval development. The authors concluded that bacteria contribute to mosquito larval development via folate biosynthesis and energy storage, and that folate supports larval development in the absence of a microbiota. The findings are sound.

I have listed a few specific comments below:

1-Line 170: In the text, it is not clear how the transcriptome analysis was validated.

Several genes that were found as the most up- and down-regulated in the transcriptomic analysis were validated via qPCR in samples obtained with the same protocol, with the only difference that the effect of the addition of folate was also investigated (now Figure S13).

We now clarify in the material and methods how qPCR expression ratios were calculated (lines 700-704).

2-Lines 193-195: "As many genes of these pathways are missing in the genome of *Ae. aegypti*, this suggests that, as reported in other insects, the mosquito microbiota participates in vitamin B". References are missing here.

We added the reference to the Kyoto encyclopaedia of gene and genomes (KEGG, lines 304-305 and 399-400)

3-Lines 197-199. "Hexamerins are proteins reported to participate in amino-acid storage during the end of larval development and metamorphosis." References are also missing here.

We added the reference to Kanost 2009 (lines 278-279).

4-Lines 321-322: "The metamorphic onset is stimulated by a peak of ecdysone which is controlled by the size and metabolic status of the larva". References are also missing here.

We added the reference to Gilbert 2000 (lines 421-423).

5- Line 341: Review written by Gondim et al. (*Insect Biochem Mol Biol.* 2018, 101:108-123) should be included in the discussion section.

We added the suggested reference (line 438).

6-Line 414: What is the composition of the fish food used for rearing mosquito larvae?

We added the specific fish food composition in the materials and methods section (lines 517-519).

7-Lines 560-570: How was the normalization of data conducted?

We added the details on how expression ratios were calculated (lines 700-704).

8-Lines 627-718. References Section contains several mistakes, typos or omissions. Some names of the journals are abbreviated, others not. Some titles of the articles are capitalized others not. Please take a close look at the Author's guides for Nature Communications.

9-Lines 632, 638, 641, 642, 643, 648, 654, 659, 662, 663, 669,672, 681, 687, 690, 695, 701: italics in scientific names are missing here.

10-Line 641: please capitalize the first letter (A) of *Aedes* and use italics for *Aedes aegypti*.

11-Line 643. Do not capitalize a of *aegypti*

12-Line 685: Why does this reference have an asterisk?

13-Line 699: The name of the journal is missing.

14-Line 706: The name of the journal is missing.

15-Line 760: A or Ae to abbreviate *Aedes*, please be consistent

We apologise for these inaccuracies, we carefully checked the whole Reference section.

16-Supplementary material:

a) Figure S1-S10: Why are the number and the title of each figure repeated twice in each figure?

We corrected these redundancies in the Supplementary Materials.

b) Figure S3: How were the data normalized? What statistical analysis was performed?

We removed this particular figure, but we now provide enough information about expression data analysis in the methods section (lines 700-704).

c) Figure S4-S6, S8: What statistical analysis was performed?

We now provide detailed information on statistical analyses in new Table S1 and lines 749-759 in the methods section.

d) Figure S9: How was the gene expression monitored? How were the data normalized? In the legend of this figure, you should also mention the Table inserted in the figure.

We now provide enough information about expression data analysis in the methods section (lines 696-698).

Comprehensive statistical information can be found in new Table S1.

e) Figure S10: Please include scale bars, and consider labeling important items.

We modified the figure (now S14) as requested.

f) Table 1S-4S. Please, provide more details. Such as how were the data obtained?

We expanded table legends to be more exhaustive. The details of the RNA Seq analysis are explained in the materials and methods section (lines 688-697).

g) Table S5: Legend of this table is incomplete.

We expanded table legends to be more exhaustive.

h) In some cases, it is indicated qPCR, and in other cases it is indicated RT-PCR. Please be consistent.

We now use the same terminology throughout the whole manuscript.

17-Supplementary references: See comments above. Please be consistent with the name of journals, titles of the articles, etc. Please use italics for scientific names of organisms.

We carefully checked and corrected the whole Reference section.

Reviewer #2 (Remarks to the Author):

In the manuscript "Bacteria contribute to mosquito development via folate biosynthesis and energy storage", Romoli et al. apply a decolonization method previously characterized in mice to mosquitos in order to generate Germ-Free (GF) adults insects. They associate GF eggs with *E. coli* HA416, which is auxotroph for two amino acids required for peptidoglycan biosynthesis. They supplement the mosquito diet with these amino acids, allowing *E. coli* colonization of larvae and thus development. When the mosquitoes reach the adult stage, or when 3rd instar larvae are transferred to a medium deprived of these amino acids, *E. coli* cannot grow anymore and the mosquitoes are rendered GF. The authors use this method to compare the transcriptomes of GF larvae and mono-associated larvae. They find differences in expression of genes involved in folate and thiamine metabolism, lipid utilization and amino acids storage. In line with previous studies, they show that folate is essential to the development of GF larvae, and that GF larvae have impaired lipid utilization.

The method presented here, which was introduced and characterized in previous works in mice, is well described and well adapted for the rapid generation of GF adults mosquitoes. This method will be useful for gnotobiotic studies on adult mosquitoes as it is probably better than the use of antibiotics. However, the main messages of the paper are not really new in the field of insect-bacteria interactions:

- The lack of GF mosquitoes has long been an issue, but as the authors cited, it was recently made possible by the work of Correa et al. where GF larvae are reared in sterile medium supplemented with an agar plug containing nutrients. In those conditions, mosquitos present a developmental delay and reduced adult size. The authors seem to consider this problematic, however stunting in GF animals has been observed in other insect models such as *Drosophila*, which suggests that animals need microbes for optimal development.
- The RNAseq results provide interesting descriptive data, but they do not lead the authors to any novel mechanism because current literature already shows the microbial contribution to folate needs of animals (Blatch et al., 2010, Piper et al., 2013, Engevik et al., 2020).
- Finally, the influence of bacteria on lipid utilization (Valzania et al., 2018) was already known.

We thank Reviewer #2 for his/her comments. We also think that this method will be useful for gnotobiotic studies on adult insects. Since the larval microbiota has an important impact on the adult physiology (Dickson et al, 2017), we believe that the strength of our method relies in the production of germ-free mosquitoes with a conventional larval development, which will allow the investigation of the role of the adult microbiota on important traits of mosquito physiology in a standardized context. Therefore, we believe that this tool is a major advance for the insect microbiota community.

We agree with reviewer #2 that the contribution of the microbiota to folate synthesis and lipid metabolism had already been observed in other models. However, the recent literature in mosquitoes proposed two alternative models (hypoxia and metabolism) which were built on well-controlled experiments and both appeared convincing to us. Thus, when we started this transcriptomic project, we did not have any strong hypothesis towards any of them. Previous transcriptomic data suffered to our eyes a problem of synchronization which is essential when dealing with development.

Major issues:

- The decolonisation method is interesting and well executed in mosquito, however it uses a gut-adapted human commensal strain. Conclusions thus cannot be extrapolated to the mosquito microbiota.

We agree that the HS strain might not be completely adapted to the mosquito gut, but *Escherichia* is one of the most frequent genera identified as a microbiota member in mosquitoes (Ramos-Nino et al, 2020; Hedge et al, 2018; Osei-Poku et al, 2012; Gendrin and Christophides, 2013). We believe that the capability of the specific HS strain to rescue larval development points out that this strain provides the metabolic support needed by the mosquito for its development.

We now provide additional CFU and development data after larval transfer (new Figure 4) showing that larvae reared in water from the field also loose a large proportion of their microbiota after being transferred to sterile water and then grows, in line with what we observed with *E. coli* and that their development success is also affected.

We discussed these results, lines 385-393.

- The title of the article does not clearly reflect the content because the main objective of the study is the adaptation and testing of the reversible colonization method.

We thank the reviewer for this comment, which illustrates that a view from outside helps providing another perspective on a study. We changed the title of the article to put more emphasis on the tool development and its implications for insect microbiota studies. The article is now entitled "*Production of germ-free mosquitoes via transient colonisation allows stage-specific investigation of host-microbiota interactions*".

- The claim in the title that "Bacteria contribute to mosquito development via folate biosynthesis and energy storage" is not well supported by the data. Gnotobiotic experiments such as colonization with WT members of the mosquito microbiota and their folate biosynthetic genes knock-out counterparts will be needed. Quantifying folate production by commensal bacteria would also help to support this claim.

We agree with Reviewer #2 that the use of bacteria deficient in folate biosynthesis, yet able to efficiently colonize larvae, would strengthen our results. However, *E. coli* mutants in these genes (e.g. *folP*, dihydropteroate synthase) need culture media rich in components whose synthesis require folate cofactors (Thymine, Glycine, Methionine, Pantothenate, Adenine or Guanine, <https://cgsc.biology.yale.edu/Site.php?ID=29566>). Since our larval rearing environment consists only of water and fish food, it would be difficult to determine if mosquito development is not rescued by these mutants because folate is absent or because bacteria are metabolically inactive. We also considered to use sulfamethoxazole, which blocks the folate pathway in bacteria by competitively inhibiting dihydropteroate synthase, preventing the synthesis of dihydrofolic acid. However, we would face the same problems mentioned above, as sulfamethoxazole has a bacteriostatic effect. Thus, we decided to change the article title and generally to tone down our statements on the contribution of the microbiota to folate biosynthesis.

- The decolonization method only produces GF 3rd instar larvae and adults. Conclusions on the impact of bacteria on development are very restricted since they are limited to the 3rd instar larvae which is a developmental stage of "preparation" to metamorphosis. Is it possible to decolonize earlier? If microbiota is important for nutrition it may also impact earlier larval stages.

If we understand well the reviewer's comment, he/she thinks that there are only three larval instars in mosquitoes, but actually there are four. We initially chose to decolonise larvae at the third instar because this stage is an intermediate between two larval stage (second and fourth instar), hence representative of larval physiology.

As suggested, we now also investigated the effect of the decolonisation at different larval stages (new Figure 3d-e, lines 193-212). We found that the production of germ-free larvae is possible at all larval stages (new Figure 3d) and that an early decolonisation affects more substantially larval development (new Figure 3e).

- Microbial impact on host physiology is dependent on active bacterial metabolism. The authors showed colonization of mosquito larvae by the bacteria tested but are bacteria actually growing and replicating in the mosquito diet upon amino acid supplementation? Or are they just persisting, alive, without doubling? It would be interesting to follow bacterial growth through time in the mosquito diet and the changes in bacterial counts upon amino acids supplementation in presence or absence of mosquito larvae.

As suggested by the reviewer, we measured bacterial counts at different time points in larval rearing water in the presence or absence of larvae. We show and describe our results in Figure S2 and lines 107-120. Briefly, bacteria do not multiply in the absence of larvae, which is likely due to quorum sensing as bacterial concentration is almost that of an overnight culture (1/5 dilution). In the presence of larvae, bacterial counts stay stable until the fourth instar, when they slightly decrease as larvae significantly grow and accumulate energy for metamorphosis.

- The authors should test whether reversible colonization allows to re-associate GF mosquito larvae with other bacteria: if there are some remaining *E. coli* AUX in the gut or in the water, they may take advantage of amino acids produced by these other bacteria to grow again. The authors should perform re-association experiments at different time points after transfer, and test for the presence of *E. coli* AUX in the gut and in the water.

As bacteria-bacteria interactions are strain specific, we think that such mutualism allowing to revive the potential last *E. coli* cells would have to be tested case by case in future recolonization studies. However, to enforce the decolonisation in adults, we now provided more qPCR data showing that bacterial DNA is not detectable in 7-to-9-day-old adults.

Minor issues:

- L46: "In the past, several studies proposed rearing protocols of germ-free larvae, which could not be reproduced in recent experiments⁸⁻¹⁰" the authors should not cite the work of Correa et al. here. It suggests that this work could not be reproduced. If that is what they actually mean, they should state it and provide references. Thanks for this point, we absolutely don't mean that Correa's work could not be reproduced!

In their work, Correa et al tried to use diets proposed by Lang et al in 1972 (referred in their paper as "synthetic larval growth medium"), which was similar to other media previously proposed by Golberg et al, 1945. However, in their hands, this diet did not succeed in rescuing larval development and larvae died. We moved the references as following to clarify this sentence:

"In the past, several studies proposed rearing protocols of germ-free larvae (Goldberg, Lang), which could not be reproduced in recent experiments (Correa)" (now lines 62-63).

- L230: "These observations suggest that bacteria are essential throughout all larval development" Is it possible to decolonize at an earlier stage?

We discussed this point above (see Major issues).

- L301: "the *E. coli* genome codes a complete biosynthetic pathway except for the gamma-glutamyl-hydrolase" the authors cite KEGG, which shows the pathway for a reference strain of *E. coli*, but strains may have different biosynthesis abilities. What about the strain used in this study and the actual mosquito microbiota strains?

This sentence specifically applies for the *E. coli* HS strain used in this study. We rephrased the sentence as following: "...the *E. coli* HS genome codes a complete biosynthetic pathway except for the gamma-glutamyl-hydrolase" (now lines 399-400).

- L302 and L314: "Our transcriptomic and diet supplementation data now provide strong evidence that bacteria and mosquito cooperate for folate biosynthesis" Can this *E. coli* strain produce THF-polyglutamate? If yes, can the authors test a knock-out mutant for THF-polyglutamate production? Can the authors provide THF-polyglutamate to the decolonized larvae and verify whether it has the same effect as folate? All these experiments are needed to support this claim. Otherwise I would recommend being more cautious in the phrasing.

We hope that the general tone of the revised manuscript (and title) will satisfy reviewer #2 as we decided not to further investigate these aspects.

- L360: Recent work shows bacterial influence on the host's expression of folate transporters (Engevik et al., 2020) did the authors find differences in regulation of folate transporters?

We thank the reviewer for this suggestion, we found indeed at least two folate transporters which expression is increased in absence of bacteria in all our samples (new Table S6, lines 306-308).

- The authors show that upon reversible colonization, there is no more culturable *E. coli* AUX at the adult stage (Fig 1D). However, they do find copies of 16S DNA (Fig S3) at D1 after eclosion. Are these copies from dead *E. coli*? Might some *E. coli* survive the lack of amino acids by entering a viable but non-culturable stage? It would be interesting to know whether these copies persist later in the adult stage.

We performed qPCR on DNA extracted from pools of midguts (sugar-fed and blood-fed mosquitoes) and pools of whole adults (new Figure 1e). Bacterial DNA was not detected in 7-to-9-day-old germ-free mosquitoes, suggesting that DNA detected at day 1 comes from DNA contamination, from dead bacteria and/or from the last persisting bacteria (2% still had some CFUs when checking a very high number of mosquitoes, Fig S3). Our new qPCR data further strengthen previous CFU data.

- Fig 4A: it is unclear to me whether the extra lipids in GF guts are in the lumen of the gut, suggesting defects in lipid uptake, or in the enterocytes, suggesting defects in lipid utilization (as it was observed in *Drosophila*, Kamareddine et al 2018).

Since midguts are dissected and undergo several washes, we believe that the lipids stained with Bodipy are stored in enterocytes (as Valzania et al and Kamareddine et al observed), suggesting a deficiency in lipid utilization. We clarified this point, lines 430-433, and discussed our results taking also into account the Kamareddine et al paper (lines 433-454).

- The authors focused their findings on the bacterial contribution to mosquito development through the regulation of host's folate metabolism, however the discussion on the role of folate in development and metabolism is scarce.

As suggested, we implemented the discussion as following: "*Folate has a central role in development and represents a key enzymatic cofactor in the one-carbon metabolism, a series of reactions controlling the production of several important nucleic acid and amino acid building blocks, such as purines, thymidine, formylated methionyl-tRNA, methionine, glycine and serine*", lines 396-399.

Reviewer #3 (Remarks to the Author):

In this manuscript, Romoli et. al. describe a novel technique to decolonize *Aedes aegypti* of their microbiome. They use a strain of *E. coli* that is auxotrophic for certain metabolites, when these are removed from the larval rearing environment the bacteria stop growing and over some time the larvae become axenic.

Developing such a system is a major step forward in the research on mosquito / insect microbiomes. I think that this methodology is well-thought out and the experiments to demonstrate how it works are well thought out.

Major points:

1. To me it is quite surprising that the authors develop a strategy to make axenic mosquitoes (that are not compromised in the way other authors have achieved this in the past) but then carry out little to no investigations on the adult stage of the mosquito. Surely this is the most relevant from many standpoints but especially from the standpoint of disease transmission. Would have been interesting to see if axenic mosquitoes generated this way had any effects on lifespan / fecundity and also to have examined gene expression in the adult stage.

We thank Reviewer #3 for his/her comments. We performed additional experiments to characterize the lifespan (new Figure 2f) and fecundity (new Figure 2d, e) of germ-free mosquitoes compared to *E. coli* WT gnotobiotic mosquitoes. These results are shown, lines 151-166, and discussed, lines 364-369. We did not investigate the expression of any gene in adults as it is outside the scope of this study. However, we found interesting that a recent transcriptomic study of germ-free adult mosquitoes obtained with the method developed by Correa et al also identified alkaline phosphatase as an important hit (Hyde et al, Sci Rep 2020) and we mentioned it in the discussion (lines 403-405).

2. The authors observe that mosquito's shed their bacteria when transferred to sterile water. It wasn't clear to me if this effect is *E. coli* specific or if the natural microbiome is also lost under these conditions. I wonder if the natural microbiome have a better ability to be retained under these conditions.

We now tested whether larvae colonised by a more native microbiota (*i.e.* reared in water collected from *Ae. aegypti* breeding sites) would lose bacteria after transfer in sterile water similarly to what observed with *E. coli* (new Figure 4). Our data show similar CFU dynamics between *E. coli* and the natural microbiota. Results are described, lines 215-237, and discussed, lines 385-393.

3. Similar to the point above, authors state on line 235, "A 60% reduction in developmental success was also observed among wild-type *E.coli* carrying larvae" - is this effect *E.coli* specific or would it also happen with the natural microbiome?

As mentioned above, larvae colonised by a native microbiota were transferred into sterile water. The developmental analysis showed a decrease in development success also for these larvae, although less marked than for *E. coli* (new Figure 4). We showed and discussed these results, lines 215-237 and 385-393.

4. Its not clear to me if the authors investigated whether Folic acid was sufficient for larval development across all instars in the absence of bacteria.

As suggested, we now checked whether folic acid alone was sufficient to rescue larval development in first-instar germ-free larvae, but this was not the case. We added these observations in text (lines 314-316 and lines 408-411) and new Figure S12.

5. Line 257-267 This section isn't well developed. It seems like a bit preliminary. Have the authors investigated the literature to determine if there is any potential link between vitamin biosynthesis and lipid transport? What about hemolymph lipid content? Is there any theory as to why without bacteria DAG and TAG cannot be generated in the enterocytes? The larval accumulation has been observed before, I wonder if there is a similar effect in adults, which has not been investigated or at least not investigated with a system with minimal limitations (reversible colonization).

We improved the discussion as suggested by the reviewer (see lines 429-449).

6. The structure of the results could be improved. For example the "gut lipid" and "hypoxia" sections are a bit lost within other results sections. While important to introduce the methods, we aren't entirely sure what the key finding of the paper is, axenic technique or folic acid?

This point was raised by another reviewer as well and helped us to think again about the presentation of our results. We now decided to put more emphasis on the method that we developed in the title and in the text.

Minor points:

1. Line 176 Important could be replaced with significant or distinct

We replaced "*important*" with "*significant*" (line 261).

2. Line 320 "cannot" is probably a bit strong in this context.

We modified the sentence as following: "*However, once they entered the metamorphosis stage, almost all pupae completed their development, suggesting that the transition between larvae and pupae is a development checkpoint while pupal development is more even*" (lines 419-421).

3. Lines 205-223 The authors spend a lot of time refuting the 'hypoxia' theory, while interesting I feel that this could be refined/ shortened.

Thanks for this comment, we shortened it (lines 284-293).

Reviewers' Comments:

Reviewer #1:

Remarks to the Author:

My questions and concerns have been adequately addressed in the revised manuscript.

Reviewer #2:

Remarks to the Author:

The authors responded to most of my concerns in a satisfying manner. They added interesting new data about natural mosquito microbiome, decolonization at early stages, as well as lifespan and fecundity measurements. Moreover, they have rephrased the manuscript so that the most important message is now the development of a new technique to generate GF adult mosquitoes. I think that it improved the clarity of the paper very much.

I still have two concerns:

- 1) I asked the authors to test *E. coli* mutants for folate synthesis. They responded that such auxotrophic mutants would not grow in the absence of folate. However, it is possible that the fish food contains traces of folate that are sufficient for *E. coli* auxotrophic mutants, especially because as shown in Fig S2A, *E. coli* does not grow much in the rearing water. This possibility was not investigated by the authors. Moreover, the authors did not try to quantify folate in the mosquito medium in absence/presence of *E. coli*. The authors decided to tune down their findings about folate to re-focus the article about the method of de-colonization, which I agree with. However, I still believe that at least one of the simple experiments suggested above would greatly improve the manuscript.
- 2) I asked whether re-association with another bacterium is possible after decolonization. The authors responded that this should be addressed by future studies, but I disagree. My concern is that if there are even a few *E. coli* AUX persisting despite the lack of amino acids, they may use amino acids produced by other bacteria to proliferate, preventing "true" mono-association. The authors show that DNA from *E. coli* AUX is not detected in 7-9 days old adults. Does it mean that future works should wait for seven days after adult emergence to perform re-association? I think that the authors should at least perform one experiment of re-association at different times after adult emergence as a proof of concept. In the future, this technique has the potential to be very useful not only to study axenic mosquitoes, but also gnotobiotic mosquitoes colonized with well-defined consortia of microbes. The authors should confirm that this is possible.

Reviewer #3:

Remarks to the Author:

The authors have done an impressive job responding to all the issue raised in my review. I think the manuscript has been improved and I have not found any further issues.

REVIEWERS' COMMENTS

Reviewer #1 (Remarks to the Author):

My questions and concerns have been adequately addressed in the revised manuscript.

We thank Reviewer #1 for his/her comments.

Reviewer #2 (Remarks to the Author):

The authors responded to most of my concerns in a satisfying manner. They added interesting new data about natural mosquito microbiome, decolonization at early stages, as well as lifespan and fecundity measurements. Moreover, they have rephrased the manuscript so that the most important message is now the development of a new technique to generate GF adult mosquitoes. I think that it improved the clarity of the paper very much.

I still have two concerns:

1) I asked the authors to test *E. coli* mutants for folate synthesis. They responded that such auxotrophic mutants would not grow in the absence of folate. However, it is possible that the fish food contains traces of folate that are sufficient for *E. coli* auxotrophic mutants, especially because as shown in Fig S2A, *E. coli* does not grow much in the rearing water. This possibility was not investigated by the authors. Moreover, the authors did not try to quantify folate in the mosquito medium in absence/presence of *E. coli*. The authors decided to tune down their findings about folate to re-focus the article about the method of de-colonization, which I agree with. However, I still believe that at least one of the simple experiments suggested above would greatly improve the manuscript.

We agree with the reviewer that such experiments would be interesting to perform and would allow us to better understand the involvement of *E. coli* in folate metabolism. However, such analyses are beyond the scope of this study.

2) I asked whether re-association with another bacterium is possible after decolonization. The authors responded that this should be addressed by future studies, but I disagree. My concern is that if there are even a few *E. coli* AUX persisting despite the lack of amino acids, they may use amino acids produced by other bacteria to proliferate, preventing "true" mono-association. The authors show that DNA from *E. coli* AUX is not detected in 7-9 days old adults. Does it mean that future works should wait for seven days after adult emergence to perform re-association? I think that the authors should at least perform one experiment of re-association at different times after adult emergence as a proof of concept. In the future, this technique has the potential to be very useful not only to study axenic mosquitoes, but also gnotobiotic mosquitoes colonized with well-defined consortia of microbes. The authors should confirm that this is possible.

We believe that even if experiments with specific microbial consortia would require seven days before re-association of germ-free mosquitoes with such bacteria, this would not be a problem as the average lifespan of mosquitoes in our setup is of 25 days for males and > 60 days for females (Figure 2).

Reviewer #3 (Remarks to the Author):

The authors have done an impressive job responding to all the issue raised in my review. I think the manuscript has been improved and I have not found any further issues.

We thank Reviewer #3 for his/her comments.